# DisCor: Corrective Feedback in Reinforcement Learning via Distribution Correction

**Aviral Kumar, Abhishek Gupta, Sergey Levine**
Electrical Engineering and Computer Sciences, UC Berkeley
`aviralk@berkeley.edu`

## Abstract

Deep reinforcement learning can learn effective policies for a wide range of tasks, but is notoriously difficult to use due to instability and sensitivity to hyperparameters. The reasons for this remain unclear. In this paper, we study how RL methods based on bootstrapping-based Q-learning can suffer from a pathological interaction between function approximation and the data distribution used to train the Q-function: with standard supervised learning, online data collection should induce corrective feedback, where new data corrects mistakes in old predictions. With dynamic programming methods like Q-learning, such feedback may be absent. This can lead to potential instability, sub-optimal convergence, and poor results when learning from noisy, sparse or delayed rewards. Based on these observations, we propose a new algorithm, DisCor, which explicitly optimizes for data distributions that can correct for accumulated errors in the value function. DisCor computes a tractable approximation to the distribution that optimally induces corrective feedback, which we show results in reweighting samples based on the estimated accuracy of their target values. Using this distribution for training, DisCor results in substantial improvements in a range of challenging RL settings, such as multi-task learning and learning from noisy reward signals.

## 1   Introduction

Reinforcement learning (RL) algorithms, when combined with high-capacity deep neural net function approximators, have shown promise in domains ranging from robotic manipulation [22] to recommender systems [44]. However, current deep RL methods can be difficult to use: they require delicate hyperparameter tuning, and exhibit inconsistent performance. While a number of hypotheses have been proposed to understand these issues [15, 52, 11, 10], and gradual improvements have led to more stable algorithms in recent years [14, 18], an effective solution has proven elusive. We hypothesize that one source of instability in reinforcement learning with function approximation and value function estimation, such as Q-learning [53, 38, 33] and actor-critic algorithms [13, 23], is a pathological interaction between the data distribution induced by the latest policy, and the errors induced in the learned approximate value function as a consequence of training on this distribution.

While a number of prior works [1, 10, 29] have provided theoretical examinations of various approximate dynamic programming (ADP) methods, which include Q-learning and actor-critic algorithms, prior work has not extensively studied the relationship between the data distribution induced by the latest value function and the *errors* in the future value functions obtained by training on this data. When using supervised learning style procedures to train contextual bandits or dynamics models, online data collection results in a kind of "hard negative" mining: the model collects transitions that lead to good outcomes according to the model (potentially erroneously). This results in collecting precisely the data needed to *correct* errors and improve. On the contrary, ADP algorithms that use

bootstrapped targets rather than ground-truth target values may not enjoy such corrective feedback with online data collection in the presence of function approximation.

Since function approximation couples Q-values at different states, the data distribution under which ADP updates are performed directly affects the learned solution. As we will argue in Section 3, online data collection may give rise to distributions that fail to correct errors in Q-values at states that are used as bootstrapping targets due to this coupling effect. If the bootstrapping targets in ADP updates are themselves are erroneous, then any form of Bellman error minimization using these targets may not result in the correction of errors in the Q-function, leading to poor performance. In this work, we show that we can explicitly address this by modifying the ADP training routine to re-weight the data buffer to a distribution that explicitly optimizes for corrective feedback, giving rise to our proposed method, **DisCor**. With DisCor, transitions sampled from the data buffer are reweighted with weights that are inversely proportional to the estimated errors in their target values. Thus, transitions with erroneous targets are down-weighted. We will show how this simple modification to ADP improve corrective feedback, and increases the efficiency and stability of ADP algorithms.

The main contribution of our work is to propose a simple modification to ADP algorithms to provide corrective feedback during the learning process, which we call DisCor. We show that DisCor can be derived from a principled objective that results in a simple algorithm that reweights the training distribution based on estimated target value error, so as to mitigate error accumulation. DisCor is general and can be used in conjunction with modern deep RL algorithms, such as DQN [33] and SAC [14]. Our experiments show that DisCor substantially improves performance of standard RL methods, especially in challenging multi-task RL settings. We evaluate our approach on both continuous control tasks and discrete-action, image-based Atari games. On the multi-task MT10 benchmark [56] and several robotic manipulation tasks, our method learns policies with a final success rate that is **50%** higher than that of SAC.

## 2   Preliminaries

The goal in reinforcement learning is to learn a policy that maximizes the expected cumulative discounted reward in a Markov decision process (MDP), which is defined by a tuple $(\mathcal{S}, \mathcal{A}, P, R, \gamma)$. $\mathcal{S}, \mathcal{A}$ represent state and action spaces, $P(s'|s, a)$ and $r(s, a)$ represent the dynamics and reward function, and $\gamma \in (0, 1)$ represents the discount factor. $\rho_0(s)$ is the initial state distribution. The infinite-horizon, discounted marginal state distribution of the policy $\pi(a|s)$ is denoted as $d^\pi(s)$ and the corresponding state-action marginal is $d^\pi(s, a) = d^\pi(s)\pi(a|s)$. We define $P^\pi$, the state-action transition matrix under a policy $\pi$ as $P^\pi Q(s, a) := \mathbb{E}_{s' \sim P(\cdot|s,a), a' \sim \pi(a'|s')}[Q(s', a')]$.

Approximate dynamic programming (ADP) algorithms, such as Q-learning and actor-critic methods, aim to acquire the optimal policy by modeling the optimal state ($V^*(s)$) and state-action ($Q^*(s, a)$) value functions by recursively iterating the Bellman optimality operator, $\mathcal{B}^*$, defined as $(\mathcal{B}^* Q)(s, a) = r(s, a) + \gamma \mathbb{E}_{s' \sim P}[\max_{a'} Q(s', a')]$. With function approximation, these algorithms project the values of the Bellman optimality operator $\mathcal{B}^*$ onto a family of Q-function approximators $\mathcal{Q}$ (e.g., deep neural nets) under a sampling or data distribution $\mu$, such that $Q_{k+1} \leftarrow \Pi_\mu(\mathcal{B}^* Q_k)$ and

$$\Pi_\mu(Q) \overset{\text{def}}{=} \arg\min_{Q' \in \mathcal{Q}} \mathbb{E}_{s,a \sim \mu}[(Q'(s, a) - Q(s, a))^2]. \tag{1}$$

Q-function fitting is usually interleaved with additional data collection, which typically uses a policy derived from the latest value function, augmented with either $\epsilon$-greedy [54, 33] or Boltzmann-style [14, 45] exploration. For commonly used ADP methods, $\mu$ simply corresponds to the on-policy state-action marginal, $\mu_k = d^{\pi_k}$ (at iteration $k$) or else a "replay buffer" [14, 33, 27, 28] formed as a mixture distribution over all past policies, such that $\mu_k = 1/k \sum_{i=1}^k d^{\pi_i}$. However, as we will show in this paper, the choice of the sampling distribution $\mu$ is of crucial importance for the stability and efficiency of ADP algorithms. We analyze this issue in Section 3, and then discuss a potential solution to this problem in Section 5.

## 3   Corrective Feedback in Q-Learning

When learning with supervised regression (i.e., non-bootstrapped objectives) onto the true value function (e.g., in a bandit setting), active data collection methods will visit precisely those state-action tuples that have erroneously optimistic values, observe their true values, and correct the errors, by fitting these true values. However, ADP methods that use bootstrapped target values may not be able to correct errors this way, and online data collection may not reduce the error between the current

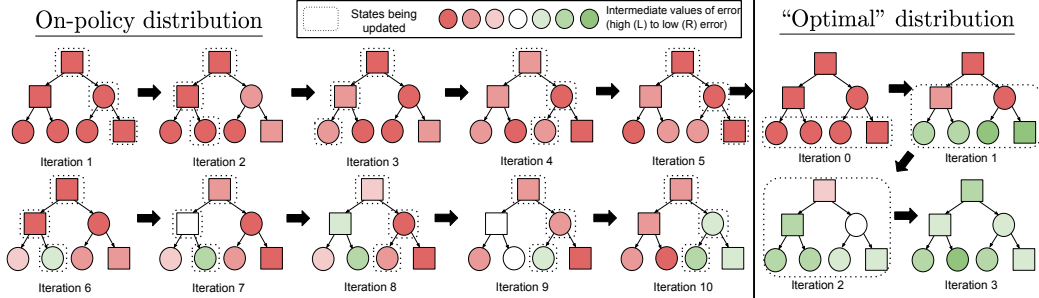

Figure 1: **Left:** Depiction of a possible run of Q-learning iterations on a tree-structured MDP with on-policy sampling. The trajectory sampled at each iteration is shown with dotted boundaries. Function approximation results in aliasing (coupling) of the box-shaped and circle-shaped nodes (i.e., instances of each shape has similar features values). Updating the values at one circle node affects all other circles, likewise for boxes. Regressing to erroneous targets at one circle node may induce errors at another circle node, *even if the other node has a correct target*, simply because the other node is visited less often. **Right:** If a distribution that puts higher probability on nodes with correct target values, i.e. which moves from leaves to nodes higher up, is chosen, then, the effects of function approximation aliasing are reduced, and correct Q-values can be obtained.

Q-function and $Q^*$, especially when function approximation is employed to represent the Q-function. This is because function approximation error can result in erroneous bootstrap target values at some state-action tuples. Visiting these tuples more often will simply cause the function approximator to more accurately fit these *incorrect* target values, rather than correcting the target values themselves. As we will show, those states that are the *cause* of incorrect target values at other states can be extremely infrequent in the data obtained by running the policy. Therefore, their values will not be corrected, leading to more error propagation.

**Didactic example.** To build intuition for the phenomenon, consider tree-structured MDP example in Figure 1. We illustrate a potential run of Q-learning (Alg. 2) with on-policy data collection. Q-values at different states are updated to match their (potentially incorrect) bootstrap target values under a distribution, $\mu(s, a)$, which, in this case is dictated by the visitation frequency under the current policy (Equation 1). The choice of $\mu(s, a)$, does not affect the resulting Q-function when function approximation is *not* used, as long as $\mu$ is full-support, i.e., $\mu(s, a) > 0 \ \forall \ s, a$.

However, with function approximation, updates across state-action pairs affect each other. Erroneous updates higher up in the tree, trying to match incorrect target values, may prevent error correction at leaf nodes if the states have similar representations under function approximation (i.e., if they are partially *aliased*). States closer to the root have higher frequencies (because there are fewer of them) than the leaves, exacerbating this problem. This issue can compound: the resulting erroneous leaf values are again used as targets for other nodes, which may have higher frequencies, further preventing the leaves from learning correct values.

If we can instead train with $\mu(s, a)$ that puts higher probability on nodes with correct target values, we can alleviate this issue. We would expect that such a method would first fit the most accurate target values (at the leaves), and only then update the nodes higher up, as shown in Figure 1 (right). Our proposed algorithm, DisCor, shows how to construct such a distribution in Section 5.

**Value error in ADP.** To more formally quantify, and devise solutions to this issue, we first define our notion of error correction in ADP in terms of *value error*:

**Definition 3.1.** *The value error is defined as the error of the current Q-function, $Q_k$ to the optimal $Q^*$ averaged under the on-policy ($\pi_k$) marginal, $d^{\pi_k}(s, a) : \mathcal{E}_k = \mathbb{E}_{d^{\pi_k}}[|Q_k - Q^*|]$.*

A smooth decrease in value error $\mathcal{E}_k$ indicates that effective error correction in the Q-function. If $\mathcal{E}_k$ fluctuates or increases, the algorithm is making poor learning progress. When the value error $\mathcal{E}_k$ is roughly stagnant at a non-zero value, this indicates premature convergence. The didactic example (Fig. 1) suggests that the value error $\mathcal{E}_k$ for ADP may not smoothly decrease to $0$, and can even increase with function approximation.

To analyze this phenomenon computationally, we use the gridworld MDPs from Fu et al. [10] and visualize the **correlations** between policy visitations $d^{\pi_k}(s, a)$ and the value of Bellman error after the ADP update, i.e. $|Q_{k+1} - \mathcal{B}^*Q_k|(s, a)$, as well as the correlation between visitations and the *difference* in value errors after and before the update, $\mathcal{E}_{k+1}(s, a) - \mathcal{E}_k(s, a)$. We eliminate finite

sampling error by training on all transitions, simply weighting them by the true on-policy or replay buffer distribution. Details are provided in Appendix G.1. In Figure 2, we show that, as expected, Bellman error correlates *negatively* with visitation frequency (dashed line), suggesting that visiting a state more often decreases its Bellman error. However, the change in value error $\mathcal{E}_{k+1} - \mathcal{E}_k$ in general *does not* correlate negatively with visitation. Value error often increases in states that are visited more frequently, suggesting that a corrective feedback mechanism is often lacking.

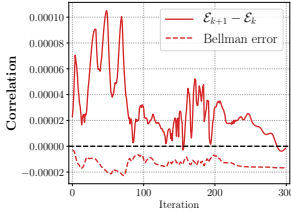

Figure 2: Correlation (y-axis) between $d^{\pi_k}(s, a)$ and the Bellman error, $|Q_{k+1} - \mathcal{B}^* Q_k|$ (dashed), and correlation between $d^{\pi_k}(s, a)$ and change in value error, $\mathcal{E}_{k+1} - \mathcal{E}_k$ (solid), during training with on-policy data. $d^{\pi_k}(s, a)$ *negatively* correlates with Bellman error, but often correlates *positively* with an increase in value error.

The Q-function value error at state-action pairs that will be used as bootstrapping targets for other state-action tuples ($Q(s_0, a_1)$ is used as target for all states with action $a_1$) is high and the state-action pair with correct target value, $(s_3, a_0)$, appears infrequently in the on-policy distribution, since the policy chooses the other action $a_1$ with high probability. Since the function approximator couples together updates across states and actions, the low update frequency at $(s_3, a_0)$ and high frequency of state-action tuples with incorrect targets will cause the Q-function updates to increase value error. Thus, minimizing Bellman error under the on-policy distribution can lead to an increase in the error against $Q^*$ (Also shown in Figure 2 on a gridworld). A more concrete computational example illustrating this phenomenon is described in detail in Section 4. We can further generalize this discussion over multiple iterations of learning.

**Which distributions lead to higher value errors?** In Figure 3, we plot value error $\mathcal{E}_k$ over the course of Q-learning with on-policy and replay buffer distributions. The plots show prolonged periods where $\mathcal{E}_k$ is increasing or fluctuating. When this happens, the policy has poor performance, with returns that are unstable or stagnating (Fig. 3). To study the effects of function approximation and distributions on this issue, we can control for both of these factors. When a uniform distribution $\text{Unif}(s, a)$ is used instead of the on-policy distribution, as shown in Fig. 3 (red), or when using a tabular representation without function approximation, but with the on-policy distribution, as shown in with Fig. 3 (brown), we see that $\mathcal{E}_k$ decreases smoothly, suggesting that the combination of function approximation and naïve distributions can result in challenges in value error reduction.

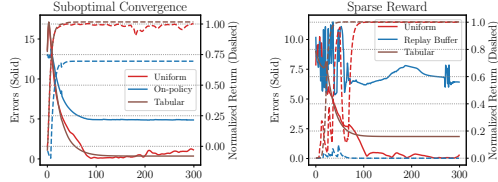

Figure 3: Value error ($\mathcal{E}_k$) and policy performance (normalized return) for **Left**: sub-optimal convergence with on-policy distributions, **Right**: instabilities in learning progress with replay buffers. Note that an oracle reweighting to a uniform data distribution or complete removal of function approximation, gives rise to decreasing $\mathcal{E}_k$ curve and better policy performance.

In fact, we can construct a family of MDPs generalizing our didactic tree example, where training with on-policy or replay buffer distributions theoretically requires at least exponentially many iterations to converge to $Q^*$, if at all convergence to $Q^*$ happens.

**Theorem 3.1** (Exponential lower bound for on-policy and replay buffer distributions)**.** *There exists a family of MDPs parameterized by $H > 0$, with $|\mathcal{S}| = 2^H$, $|\mathcal{A}| = 2$ and state features $\Phi$, such that on-policy or replay-buffer Q-learning requires $\Omega\left(\gamma^{-H}\right)$ exact Bellman projection steps for convergence to $Q^*$, if at all convergence happens. This happens even with features, $\Phi$ that can represent the optimal Q-function near-perfectly, i.e., $||Q^* - \Phi w||_\infty \leq \varepsilon$.*

The proof is in Appendix D. This suggests that on-policy or replay buffer distributions can induce very slow learning in certain MDPs. We show in Appendix D.3 that our method, DisCor, which we derive in the next section, can avoid many of these challenges, in this MDP family.

## 4 Optimal Distributions for Value Error Reduction

We discussed how, with function approximation and on-policy or replay-buffer training distributions, the value error $\mathcal{E}_k$ may not decrease over the course of training. What if we instead directly optimize the data distribution at each iteration so as to minimize value error? To do so, we derive a functional form for this "optimal" distribution by formulating an optimization problem that directly optimizes the training distribution $p_k(s, a)$ at each iteration $k$, greedily minimizing the error $\mathcal{E}_k$ at the end of

iteration $k$. Note that $p_k(s,a)$ is now distinct from the on-policy or buffer data distribution denoted by $\mu(s,a)$. We will then show how to approximately solve for $p_k(s,a)$, yielding a simple practical algorithm in Section 5. All proofs are in Appendix A. We can write the optimal $p_k(s,a)$ as the solution to the following optimization problem:

$$\min_{p_k} \ \mathbb{E}_{d^{\pi_k}}\left[|Q_k - Q^*|\right] \ \text{s.t.} \ Q_k = \arg\min_Q \mathbb{E}_{p_k}\left[(Q - \mathcal{B}^*Q_{k-1})^2\right], \quad \sum_{s,a} p_k(s,a) = 1. \quad (2)$$

**Theorem 4.1.** *The solution $p_k(s,a)$ to a relaxation of the optimization in Equation 2 satisfies*

$$p_k(s,a) \propto \exp\left(-|Q_k - Q^*|(s,a)\right) \frac{|Q_k - \mathcal{B}^*Q_{k-1}|(s,a)}{\lambda^*}, \quad (3)$$

*where $\lambda^* \in \mathbb{R}^+$ is the magnitude of Lagrange multiplier for $\sum_{s,a} p_k(s,a) = 1$ in Problem 2.*

**Proof sketch.** Our proof of Theorem 4.1 utilizes the Fenchel-Young inequality [39] to first upper-bound $\mathbb{E}_{d^{\pi_k}}\left[|Q_k - Q^*|\right]$ via more tractable terms giving us the relaxation, and then optimizing the Lagrangian. We We use the implicit function theorem (IFT) [24] to compute implicit gradients of $Q_k$ with respect to $p_k$.

Intuitively, the optimal $p_k$ in Equation 3 assigns higher probability to state-action tuples with high Bellman error $|Q_k - \mathcal{B}^*Q_{k-1}|$, but only when the resulting Q-value $Q_k$ is close to $Q^*$. However, this expression contains terms that depend on $Q^*$ and $Q_k$, namely $|Q_k - Q^*|$ and $|Q_k - \mathcal{B}^*Q_{k-1}|$, which are observed only *after* $p_k$ is chosen. As we will show next, we need to estimate these quantities using surrogates, that only depend upon the past Q-function iterates in order to use $p_k$ in a practical algorithm. Intuitively, these surrogates exploit the rich structure in Bellman iterations: the Bellman error at *each* iteration contributes to the error against $Q^*$ in a structured manner, as we will discuss below, allowing us to approximate the value error using a special sum of Bellman errors. We present these approximations below, and then combine then to derive our proposed algorithm, DisCor.

**Surrogate for $|Q_k - Q^*|$.** For approximating the error against $Q^*$, we show that the cumulative sum of discounted and propagated Bellman errors over the past iterations of training, denoted as $\Delta_k$ and shown in Equation 5, are equivalent to an upper bound on $|Q_k - Q^*|$. Specifically, Theorem 4.2 will show that, up to a constant, $\Delta_k$ forms a tractable upper bound on $|Q_k - Q^*|$ constructed only from prior Q-function iterates, $Q_0, \cdots, Q_{k-1}$. We define $\Delta_k$ as:

$$\Delta_k = \sum_{i=1}^{k} \gamma^{k-i} \left(\prod_{j=i}^{k-1} P^{\pi_j}\right) |Q_i - (\mathcal{B}^*Q_{i-1})|. \quad \text{(vector-matrix form of}\Delta) \quad (4)$$

$$\implies \Delta_k(s,a) = |Q_k(s,a) - (\mathcal{B}^*Q_{k-1})(s,a)| + \gamma(P^{\pi_{k-1}}\Delta_{k-1})(s,a). \quad (5)$$

Here $P^{\pi_j}$ is the state-action transition matrix under policy $\pi_j$ as described in Section 2. We can then use $\Delta_k$ to define an upper bound on the value error $|Q_k - Q^*|$, as follows:

**Theorem 4.2.** *There exists a $k_0 \in \mathbb{N}$, such that $\forall \ k \geq k_0$ and $\Delta_k$ from Equation 5, $\Delta_k$ satisfies the following inequality, pointwise, for each $s,a$, as well as, $\Delta_k \to |Q_k - Q^*|$ as $\pi_k \to \pi^*$.*

$$\Delta_k(s,a) + \sum_{i=1}^{k} \gamma^{k-i}\alpha_i \geq |Q_k - Q^*|(s,a), \ \alpha_i = \frac{2R_{\max}}{1-\gamma}D_{\mathrm{TV}}(\pi_i(\cdot|s), \pi^*(\cdot|s)).$$

A proof and intermediate steps of simplification can be found in Appendix B. The key insight in this argument is to use a recursive inequality, presented in Lemma B.1, App. B, to decompose $|Q_k - Q^*|$, which allows us to show that $\Delta_k + \sum_i \gamma^{k-i}\alpha_i$ is a solution to the corresponding recursive equality, and hence, an upper bound on $|Q_k - Q^*|$. Using an upper bound of this form in Equation 3 may downweight more transitions, but will never upweight a transition that should not be upweighted.

**Estimating $|Q_k - \mathcal{B}^*Q_{k-1}|$.** The Bellman error multiplier term $|Q_k - \mathcal{B}^*Q_{k-1}|$ in Equation 3 is also not known in advance. Since no information is known about the Q-function $Q_k$, a viable approximation is to bound $|Q_k - \mathcal{B}^*Q_{k-1}|$ between the minimum and maximum Bellman errors obtained at the previous iteration, $c_1 = \min_{s,a}|Q_{k-1} - \mathcal{B}^*Q_{k-2}|$ and $c_2 = \max_{s,a}|Q_{k-1} - \mathcal{B}^*Q_{k-2}|$. We restrict the support of state-action pairs $(s,a)$ used to compute $c_1$ and $c_2$ to be the set of transitions in the replay buffer used for the Q-function update, to ensure that both $c_1$ and $c_2$ are finite. This

approximation can then be applied to the solution obtained in Equation 3 to replace the Bellman error multiplier $|Q_k - \mathcal{B}^* Q_{k-1}|$, effectively giving us a lower-bound on $p_k(s, a)$ in terms of $c_1$ and $c_2$.

**Re-weighting the replay buffer** $\mu$. Since it is challenging to directly obtain samples from $p_k$ via online interaction, a practically viable alternative is to use the samples from a standard replay buffer distribution, denoted $\mu$, but reweight these samples using importance weights $w_k = p_k(s, a)/\mu(s, a)$. However, naïve importance sampling often suffers from high variance, leading to unstable learning. Instead of directly re-weighting to $p_k$, we re-weight samples from $\mu$ to a projection of $p_k$, denoted as $q_k$, that is still close to $\mu$ under the KL-divergence metric, such that $q_k = \arg\min_q \mathbb{E}_{q(s,a)}[\log p_k(s, a)] + \tau \mathrm{D}_{\mathrm{KL}}(q(s, a)||\mu(s, a))$, where $\tau > 0$ is a scalar. The weights $w_k$ are thus given by (derivation in Appendix B):

$$ w_k(s, a) \propto \exp\left(\frac{-|Q_k - Q^*|(s, a)}{\tau}\right) \frac{|Q_k - \mathcal{B}^* Q_{k-1}|(s, a)}{\lambda^*} \tag{6} $$

**Putting it all together.** We have noted all practical approximations to the expression for optimal $p_k$ (Equation 3), including estimating surrogates for $Q_k$ and $Q^*$, and the usage of importance weights to simply *re-weighting transitions* in the replay buffer, rather than altering the exploration strategy. We now put these together to obtain a tractable expression for weights in our method. Due to space limitations, we only provide a sketch of the proof here, and a detailed derivation is in Appendix C.

We first upper-bound the quantity $|Q_k - Q^*|$ by $\Delta_k$. However, estimating $\Delta_k$ requires $|Q_k - \mathcal{B}^* Q_{k-1}|$, which is not known in advance. We utilize the upper bound $c_2$: $|Q_k - \mathcal{B}^* Q_{k-1}|(s, a) \leq c_2$, and hence use $\gamma P^{\pi_{k-1}} \Delta_{k-1}(s, a) + c_2$ as an estimator for $|Q_k - Q^*|$ in Equation 6. For the final Bellman error term outside the exponent, we can lower bound it with $c_1$, where $|Q_k - \mathcal{B}^* Q_{k-1}| \geq c_1$. Simplifying constants $c_1$, $c_2$ and $\lambda^*$, the final expression for this tractable approximation for $w_k$ is:

$$ w_k(s, a) \propto \exp\left(-\frac{\gamma\left[P^{\pi_{k-1}} \Delta_{k-1}\right](s, a)}{\tau}\right). \tag{7} $$

This expression gives rise to our practical algorithm, DisCor, described in the next section.

**A concrete demonstration.** To illustrate the effectiveness of DisCor and the challenges with naively chosen distributions in RL, we present a simple computational example in Figure 4 that illustrates that, even in a simple MDP, error can increase with standard Q-learning but decreases with our distribution correction approach, DisCor, that is based on the idea of first attempting to minimize value error at states-action tuples that will serve as target-values for other states. Our example is a 5-state MDP, with the starting state $s_0$ and the terminal state $s_T$ (marked in gray). Each state has two available actions, $a_0$ and $a_1$, and each action deterministically transits the agent to a state marked by arrows in Figure 4. A reward of 0.001 is received only when action $a_0$ is chosen at state $s_3$ (else reward is 0). The Q-function is a linear function over pre-defined features $\phi(s, a)$, i.e., $Q(s, a) = [w_1, w_2]^T \phi(s, a)$, where $\phi(\cdot, a_0) = [1, 1]$ and $\phi(\cdot, a_1) = [1, 1.001]$ (hence features are aliased across states). Computationally, we see that when minimizing Bellman error starting from a Q-function with weights $[w_1, w_2] = [0, 1e\text{-}4]$, under the on-policy distribution of the Boltzmann policy, $\pi(a_0|\cdot) = 0.001, \pi(a_1|\cdot) = 0.999$, in the absence of sampling error (using all transitions but weighted), the error against $Q^*$ still **increases** from 7.177e-3 to 7.179e-3 in one iteration, whereas with DisCor error **decreases** to **5.061e-4**. With uniform the error also decreases, but is larger: 4.776e-3.

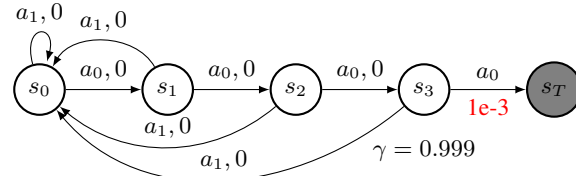

Figure 4: A simple MDP showing the effect of on-policy distribution and function approximation on learning dynamics of ADP algorithms.

## 5 Distribution Correction (DisCor) Algorithm

In this section, we present the our full, practical algorithm, which uses the weights $w_k$ from Equation 7 to re-weight the Bellman backup in order to better correct value errors. Pseudocode for our approach, called **DisCor** (**Dis**tribution **Cor**rection), is presented in Algorithm 1, with the main differences from standard ADP methods highlighted in red. In addition to a standard Q-function, DisCor trains another parametric model, $\Delta_\phi$, to estimate $\Delta_k(s, a)$ at each state-action pair. The recursion in Equation 5 is used to obtain a simple approximate dynamic programming update rule for the parameters $\phi$ (Line

8). We need to explicitly estimate this error term $\Delta_\phi$ because it is required to compute the weights described in Equation 7. The second change is the introduction of a weighted Q-function backup with weights $w_k(s, a)$, as shown in Equation 7 on Line 7. Since DisCor simply introduces a change to the training distribution, this change can be applied to popular ADP algorithms such as DQN [33] or SAC [14], as shown in Algorithm 3, Appendix F.

Using the weights $w_k$ in Equation 7 for weighting Bellman backups possesses a very clear and intuitive explanation. Note that $(P^{\pi_{k-1}}\Delta_{k-1})(s, a)$ corresponds to the estimated upper bound on the error of the target values for the current transition, due to the backup operator $P^{\pi_{k-1}}$, as described in Equation 7. Intuitively, this implies that weights $w_k$ *downweight* those transitions for which the bootstrapped *target* Q-value estimate has a high estimated error to $Q^*$, effectively focusing the learning on samples where the supervision (target value) is estimated to be accurate, which are precisely the samples that we expect maximally improve the accuracy of the Q function.

---

**Algorithm 1 DisCor (Distribution Correction)**

1: Initialize Q-values $Q_\theta(s, a)$, initial distribution $p_0(s, a)$, a replay buffer $\mu$, and an error model $\Delta_\phi(s, a)$.
2: **for** step $k$ in $\{1, \ldots, N\}$ **do**
3:    Collect $M$ samples using $\pi_k$, add them to replay buffer $\mu$, sample $\{(s_i, a_i)\}_{i=1}^N \sim \mu$
4:    Evaluate $Q_\theta(s, a)$ and $\Delta_\phi(s, a)$ on samples $(s_i, a_i)$.
5:    Compute target values for $Q$ and $\Delta$ on samples:
      $y_i = r_i + \gamma \max_{a'} Q_{k-1}(s'_i, a')$
      $\hat{a}_i = \arg\max_a Q_{k-1}(s'_i, a)$
      $\hat{\Delta}_i = |Q_\theta(s, a) - y_i| + \gamma \Delta_{k-1}(s'_i, \hat{a}_i)$
6:    Compute $w_k$ using Equation 7.
7:    Minimize Bellman error for $Q_\theta$ weighted by $w_k$.
      $\theta_{k+1} \leftarrow \underset{\theta}{\arg\min} \frac{1}{N} \sum_i^N w_k(s_i, a_i)(Q_\theta(s_i, a_i) - y_i)^2$
8:    Minimize ADP error for training $\phi$.
      $\phi_{k+1} \leftarrow \underset{\phi}{\arg\min} \frac{1}{N} \sum_{i=1}^N (\Delta_\phi(s_i, a_i) - \hat{\Delta}_i)^2$
9: **end for**

---

# 6 Related Work

Prior work has pointed out a number of issues arising when dynamic programming is used with function approximation. [35, 36, 8, 43, 26, 42] focused on analysing error induced in Bellman projections, under the assumption of an abstract error model. Convergent backups [47, 46, 32] were developed. However, divergence is rarely observed to be an issue with deep Q-learning methods [10, 52]. In contrast to these works, which mostly focus on convergence of the Bellman backup, we focus on the interaction between the ADP update and the data distribution $\mu$. Prior work on Q-learning and stochastic approximation analyzes time-varying $\mu$, but either without function approximation [53, 49, 5], or when fully online [50], unlike our setting, that uses replay buffer data.

While generalization effects of deep neural nets with ADP updates have been studied [1, 10, 30, 25], often under standard NTK [21] assumptions [1], the high-level idea in these prior works has been to suppress any coupling effects of the function approximator, effectively obtaining tabular behavior. In contrast, DisCor solves an optimization problem for the distribution $p_k$ that maximally reduces value error, and does not explicitly suppress coupling effects, as these can be important for generalization in high dimensions. [41] studies the effect of data distribution on multi-objective policy gradient methods and reports a pathological interaction between the data distribution and optimization. [9] shows the existence of suboptimal fixed points with on-policy TD learning as we observed empirically in Figure 3 (left). DisCor re-weights the transition in the buffer based on an estimate of their error to the true optimal value function. This scheme resembles learning with noisy labels via "abstention" from training on labels that are likely to be inaccurate [48]. Prioritized sampling has been used previously in ADP methods to instead prioritize transitions with higher Bellman error [40, 17, 20, 19]. We show in Section 7 that this approach is less effective than DisCor experimentally. Recent work [6] has attempted to use a distribution-checking oracle to control the amount of exploration performed. DisCor, instead, re-weights the data distribution without requiring any oracles.

# 7 Experimental Evaluation of DisCor

The goal of our empirical evaluation is to study the following questions: **(1)** Does DisCor lead to a decrease in value error, mitigating the issues raised in Section 3?, **(2)** How do approximations from Section 4 affect the efficacy of DisCor in ensuring value error reduction? **(3)** How does DisCor compare to prior methods, including those that also reweight the data in various ways?, **(4)** Can DisCor attain good performance in challenging settings, such as multi-task RL, robotic manipulation or Atari games? We start by presenting an analysis on tabular MDPs with function approximation, and then study six robotic manipulation tasks and multi-task RL, and three Atari games [3].

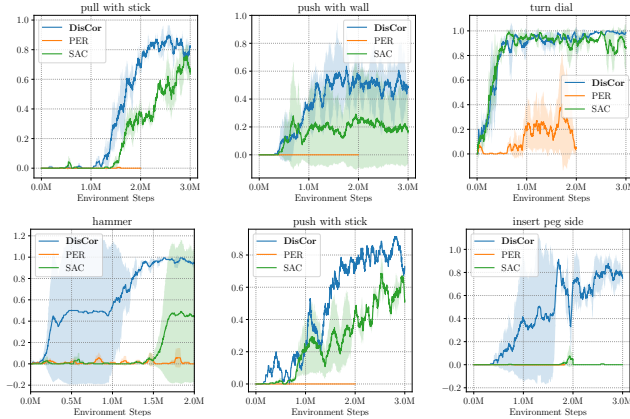

Figure 5: Evaluation success of DisCor, unweighted SAC and PER on six MetaWorld tasks. From left to right: pull stick, push with wall, push stick, turn dial, hammer and insert peg side. Note that DisCor achieves better final success rates or learns faster on most of the tasks and is the only method that learns on one task.

## 7.1 Analysis of DisCor on Tabular Environments

We first use the tabular domains from Section 3, described in detail in Appendix G.1, to analyze error correction induced by DisCor and evaluate the effect of the approximations used in our method, such as the upper bound estimator $\Delta_k$, both when the algorithm is provided with Fig. 12). all transitions in the replay buffer and simply chooses a weighting on them (no sampling error) and when the algorithm collects its own transitions via exploration. In both settings, in Figure 6, value error $\mathcal{E}_k$ decreases smoothly with DisCor. An oracle version of the algorithm (DisCor (oracle); Equation 6), which uses the true error $|Q_k - Q^*|$ in place of $\Delta_k$, is somewhat better than DisCor (Fig. 7, red vs blue), but DisCor still outperforms on-policy and replay buffer schemes (green and pink), which often fail to reduce $\mathcal{E}_k$ as shown in Section 3. While Dis-Cor (oracle) consistently performs better than DisCor, as we would expect, the approximate DisCor algorithm still attains better performance than naïve uniform weighting and prioritization similar to PER. This shows that the principle behind DisCor is effective when applied exactly, and that even the approximation that we use in practice improves performance.

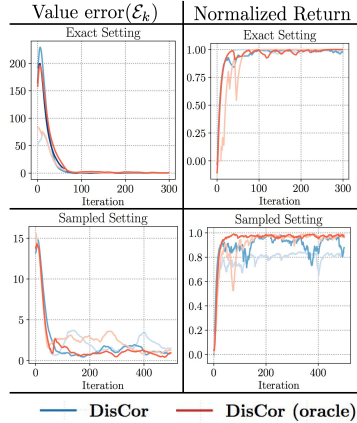

Figure 6: Value Error $\mathcal{E}_k$/ return for *two* runs of DisCor (blue) and DisCor (oracle) (red) in exact (top) and sampled (bottom) settings. Note (i) DisCor achieves similar performance as DisCor (oracle), (ii) $\mathcal{E}_k$ generally decreases with both methods.

## 7.2 Continuous Control Experiments

We next perform a comparative evaluation of DisCor on several continuous control tasks, using six robotic manipulation tasks from the Meta-World suite (pull stick, hammer, insert peg side, push stick, push with wall and turn dial) (these are shown in Figure 15 in Appendix G). These domains were chosen because they are challenging for state-of-the-art RL methods, such as SAC [14]. We applied DisCor to these tasks by modifying the weighting of samples in SAC. DisCor does not alter any hyperparameter from SAC, and requires minimal tuning. There is only one additional temperature hyperparameter, which is also automatically chosen. More details are presented in Appendix F.2.

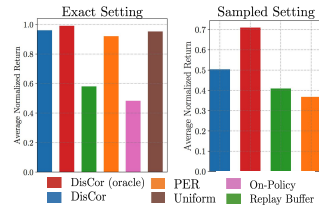

Figure 7: Performance of **DisCor**, **DisCor (oracle)** and other distributions averaged across tabular domains with and without sampling error.. DisCor is generally comparable to DisCor (oracle), and both of them generally outperform all other distributions.

We compare DisCor to standard SAC without weighting, as well as prioritized experience replay (PER) [40], which uses weights based on the last Bellman error. The results in Figure 5 show that DisCor outperforms prior methods on these tasks. DisCor learns substantially faster on most of the tasks. We also performed comparisons on the more conventional gym benchmarks, where we see a small but consistent benefit from DisCor reweighting. Since prior methods, such as SAC already

solve these tasks easily, and have been tuned well for them, the room for improvement is very small. We include these results in Appendix G.3 for completeness. We also evaluate on a stochastic reward variant of gym, where we observe an improvement trend. However, on tasks that have not been tuned as extensively or exhibit challenging properties, such as multi-task learning or complex manipulation tasks, current RL methods can perform poorly.

## 7.3  Multi-Task Reinforcement Learning

Another challenging setting for current RL methods is the multi-task RL setting. This is known to be difficult, to the point that often times learning completely separate policies for each of the tasks is actually faster, and results in better performance, than learning the tasks together [56, 41]. We evaluate on the MT10 MetaWorld benchmark [56], which consists of ten robotic manipulation tasks to be learned jointly. We follow the proto-

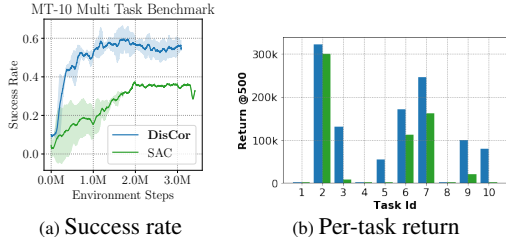

(a) Success rate                    (b) Per-task return

Figure 8: Performance of DisCor (blue) and unweighted SAC (green) on the MT10 benchmark. We observe that: (1) DisCor outperforms unweighted SAC by a factor of **1.5** in terms success rate; (2) DisCor achieves a non-trivial return on **7/10** tasks after 500k environment steps, as compared to **3/10** for unweighted SAC.

col from [56], and append task ID to the state. As shown in Figure 8(a), DisCor outperforms SAC by a large margin, achieving **50%** higher success rates compared to SAC, and a high overall return (Fig 18). Figure 8(b) shows that DisCor makes progress on **7/10** tasks, as compared to **3/10** for SAC. We further evaluate DisCor and SAC on the more challenging MT50 benchmark [56], shown in Figure 19, and observe a similar benefit as compared to MT10, where the baseline algorithm tends to plateau suboptimally for about 4M environment steps, whereas DisCor keeps learning, and achieves asymptotic performance faster.

## 7.4  Arcade Learning Environment

Our final experiments were aimed at testing the efficacy of DisCor on stochastic, discrete-action, image-observation environments. To this end, we evaluated DisCor on three commonly reported tasks from the Atari suite – Pong, Breakout and Asterix. We compare to the base-line DQN [33], all our implementations are built off of Dopamine [4], and use the evaluation protocol

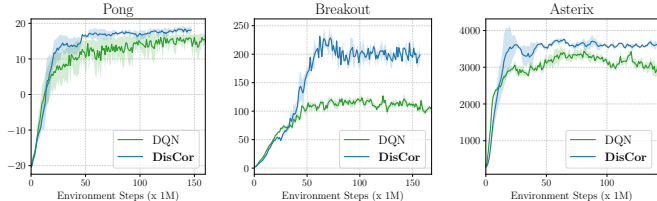

Figure 9: DQN vs DisCor on Atari. Note that DisCor generally improves learning speed and asymptotic performance.

with sticky actions [31]. We build DisCor on top of DQN by simply replacing the standard replay buffer sampling scheme in DQN with the DisCor weighted update. We show in Figure 9 that DisCor usually outperforms unweighted DQN in learning speed and performance.

# 8  Discussion, Future Work and Open Problems

In this work, we show that deep RL algorithms are unable to correct errors in the value function in scenarios with naïve online data collection. This results in a number of problems during learning, including slow convergence, inability to convergence and oscillation. We propose a method to compute the optimal data distribution to obtain value error correction, and design a practical algorithm, DisCor, that applies this correction by re-weighting the transitions in the replay buffer based on an estimate of the accuracy of their target values. DisCor yields improvements across a wide range of RL problems, including challenging robotic manipulation tasks, multi-task reinforcement learning and Atari games and can be easily combined with a variety of ADP algorithms.

This suggests several exciting directions for future work. First, a characterization of the learning dynamics and their interaction with corrective feedback and data distributions in ADP algorithms will lead to even better and more stable algorithms. Second, we could study how we might directly modify the exploration policy to change which transitions are collected, so as to more directly alter the training distribution. Third, the general theme of re-weighting and reorganizing experience with the goal of significantly simplifying optimization in ADP methods is likely to be fruitful in devising better algorithms. If we can devise RL methods that are guaranteed to enjoy corrective feedback then RL algorithms can be reliably scaled to large open-world settings.

## Broader Impact

Approximate dynamic programming methods are a key ingredient in modern deep reinforcement learning algorithms, which have had successes on a number of practical problems. However, reinforcement learning algorithms are still limited by problems such as instability and sensitivity to hyperparameters. In this work, we analyzed one such issue, which we called "corrective feedback", that afflicts modern ADP algorithms: the interaction between online data distributions and function approximation may not be able to correct errors in the Q-function. By optimizing for the distribution to maximize corrective feedback, our proposed method, DisCor, significantly improves RL problems in several challenging reinforcement learning and multi-task reinforcement learning settings, across a wide range of domains. DisCor is simple, intuitive, and admits a principled derivation, and can be applied with a number of modern deep RL algorithms.

The broad theme behind this work is to identify and address problems that arise with deep reinforcement learning algorithms. Reinforcement learning algorithms often enjoy provable guarantees with tabular representations, but it is unclear how to extend these to deep networks. Instead we could take an alternate approach: optimize for quantities of interest (such as in our case, corrective feedback) directly. We believe that this general principle can help scale end-to-end learning autonomous decision-making based approaches to real-life problems, such as robotics, software systems and autonomous driving.

Machines that can reason and perform autonomous decision-making have a wide range of applications, in a wide range of domains, and like any other technological innovations that mankind has seen, effective autonomous decision-making has both positive and negative societal effects. While effective autonomous decision-making can have considerable positive economic effects, such as by automating manufacturing lines, and other positive effects, that enhance human life quality, it can have complex economic effects due to changing economic conditions (e.g., changing job requirements, loss of jobs in some sectors and growth in others, etc.). Such implications apply broadly to technologies that enable automation agnostic of data-driven learning or reinforcement learning, and are largely not unique to this specific work.

## Acknowledgements and Funding Disclosures

We thank Xinyang Geng and Aurick Zhou for helpful discussions. We thank Vitchyr Pong, Greg Kahn, Xinyang Geng, Aurick Zhou, Avi Singh, Nicholas Rhinehart, and Michael Janner for feedback on an earlier version of this paper, and all the members of the RAIL lab for their help and support. We thank Tianhe Yu, Kristian Hartikainen, and Justin Yu for help with debugging and setting up various tasks and implementations. This research was supported by: the National Science Foundation, the Office of Naval Research, and the DARPA Assured Autonomy program. We thank Google, Amazon and NVIDIA for providing compute resources.

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
