[Supplementary Material]

# Appendices

## A Detailed Proof of Theorem 4.1 (Section 4)

In this appendix, we present a detailed proofs for the theoretical derivation of DisCor outlined in Section 4. To get started, we mention the optimization problem being solved for convenience.

$$\min_{p_k} \ \mathbb{E}_{d^{\pi_k}}\left[|Q_k - Q^*|\right]$$
$$\text{s.t.} \ Q_k = \arg\min_Q \mathbb{E}_{p_k}\left[(Q - \mathcal{B}^* Q_{k-1})^2\right]. \tag{8}$$

We break down this derivation in steps marked as relevant paragraphs. The first step is to decompose the objective into a more tractable one via an application of the Fenchel-Young inequality [39].

**Step 1: Fenchel-Young Inequality.** The optimization objective in Problem 8 is the inner product of $d^{\pi_{k-1}}$ and $|Q_k - Q^*|$. We can decompose this objective by applying the Fenchel-Young inequality [39]. For any two vectors, $x, y \in \mathbb{R}^d$, and any convex function $f$ and its Fenchel conjugate $f^*$, we have that, $x^T y \leq f(x) + f^*(y)$. We therefore have:

$$\mathbb{E}_{d^{\pi_k}}\left[|Q_k - Q^*|\right] \leq f\left(|Q_k - Q^*|\right) + f^*\left(d^{\pi_k}\right). \tag{9}$$

Since both sides of the above equation have the same minimum (here the minima are given by $Q_k = Q^*$), we can replace the objective in Problem 8 with the upper bound in Equation 9 and solve the relaxed optimization problem. As we will see below, a convenient choice of $f$ is the *soft-min* function:

$$f(x) = -\log\left(\sum_i e^{-x_i}\right), \ f^*(y) = \mathcal{H}(y). \tag{10}$$

$f^*$ in this case is given by the Shannon entropy, which is defined as $\mathcal{H}(y) = -\sum_j y_j \log y_j$. Plugging this back in problem 8, we obtain an objective that dictates minimization of the marginal state-action entropy of the policy $\pi$.

In order to make this objective even more convenient and tractable, we upper bound the Shannon entropy, $\mathcal{H}(y)$ by the entropy of the uniform distribution over states and actions, $\mathcal{H}(\mathcal{U})$. This step ensures that the entropy of the state-action marginal $d^\pi$ is not reduced drastically due to the choice of $p$. We can now minimize this upper bound, since minimizing an upper bound, leads to a minimization of the original problem, and therefore, we obtain the following new optimization problem shown in Equation 11 is:

$$\min_{p_k} \ -\log\left(\sum_{s,a} \exp(-|Q_k - Q^*|(s,a))\right)$$
$$\text{s.t.} \ Q_k = \arg\min_Q \mathbb{E}_{p_k}\left[(Q - \mathcal{B}^* Q_{k-1})^2\right]. \tag{11}$$

Another way to interpret this step is to modify the objective in Problem 8 to maximize entropy-augmented objective as is common in a number of prior works, albeit with entropy over different distributions such as [16, 14]. This also increases the smoothness of the loss landscape, which is crucial for performance of RL algorithms [2].

**Step 2: Computing the Lagrangian.** In order to solve optimization problem Problem 11, we follow standard procedures for finding solutions to constrained optimization problems. We first write the Lagrangian for this problem, which includes additional constraints to ensure that $p_k$ is a valid distribution:

$$\mathcal{L}(p_k; \lambda, \mu) = -\log\left(\sum_{s,a} \exp(-|Q_k - Q^*|(s,a))\right) + \lambda\left(\sum_{s,a} p_k(s,a) - 1\right) - \mu^T p_k. \tag{12}$$

with constraints $\sum_{s,a} p_k(s,a) = 1$ and $p_k(s,a) \geq 0$ ($\forall s, a$) and their corresponding Lagrange multipliers, $\lambda$ and $\mu$, respectively, that ensure $p_k$ is a valid distribution. An optimal $p_k$ is obtained by setting the gradient of the Lagrangian with respect to $p_k$ to 0. This requires computing the gradient of $Q_k$, resulting from Bellman error minimization, i.e. computing the derivative through the solution of another optimization problem, with respect to the distribution $p_k$. We use the implicit function theorem (IFT) [24] to compute this gradient. We next present an application of IFT in our scenario.

**Step 3: IFT gradient used in the Lagrangian.** We derive an expression for $\frac{\partial Q_k}{\partial p_k}$ which will be used while computing the gradient of the Lagrangian Equation 12 which involves an application of the implicit function theorem. The IFT gradient is given by:

$$\left.\frac{\partial Q_k}{\partial p_k}\right|_{Q_k, p_k} = -\left[\mathrm{Diag}(p_k)\right]^{-1}\left[\mathrm{Diag}(Q_k - \mathcal{B}^*Q_{k-1})\right]. \tag{13}$$

To get started towards showing Equation 13, we consider a non-parametric representation for $Q_k$ (a table), so that we can compute a tractable term without going onto the specific calculations for Jacobian or inverse-Hessian vector products for different parametric models. In this case, the Hessians in the expression for IFT and hence, the implicit gradient are given by:

$$H_Q = 2\,\mathrm{Diag}(p_k) \qquad H_{Q,p_k} = 2\,\mathrm{Diag}(Q_k - \mathcal{B}^*Q_{k-1})$$
$$\frac{\partial Q_k}{\partial p_k} = -\left[H_Q\right]^{-1}H_{Q,p_k} = -\mathrm{Diag}\left(\frac{Q_k - \mathcal{B}^*Q_{k-1}}{p_k}\right) \qquad . \tag{14}$$

provided $p_k \geq 0$ (which is true, since we operate in a full coverage regime, as there is no exploration bottleneck when all transitions are provided). This quantity is $0$ only if the Bellman residuals $Q_k - \mathcal{B}^*Q_{k-1}$ are $0$, however, that is rarely the case, hence this gradient is non-zero, and intuitively quantifies the right relationship: Bellman residual errors $Q_k - \mathcal{B}^*Q_{k-1}$ should be higher at state-action pairs with low density $p_k$, indicating a roughly inverse relationship between the two terms – which is captured by the IFT term.

**Step 4: Computing optimal $p_k$.** Now that we have the equation for IFT (Equation 13) and an expression for the Lagrangian (Equation 12), we are ready to compute the optimal $p_k$ via an application of the KKT conditions. At an optimal $p_k$, we have,

$$\frac{\partial \mathcal{L}(p_k; \lambda, \mu)}{\partial p_k} = 0 \implies \frac{\mathrm{sgn}(Q_k - Q^*)\exp(-|Q_k - Q^*|(s, a))}{\sum_{s',a'}\exp(-|Q_k - Q^*|(s', a'))} \cdot \frac{\partial Q_k}{\partial p_k} + \lambda - \mu_{s,a} = 0. \tag{15}$$

Now, re-arranging Equation 15 and plugging in the expression for $\frac{\partial Q_k}{\partial p_k}$ from Equation 13 in this Equation to obtain an expression for $p_k(s, a)$, we get:

$$p_k(s, a) \propto \exp\left(-|Q_k - Q^*|(s, a)\right)\frac{|Q_k - \mathcal{B}^*Q_{k-1}|(s, a)}{\lambda^*}. \tag{16}$$

Provided, each component of $p$ is positive, i.e. $p_k(s, a) \geq 0$ for all $s, a$, the optimal dual variable $\mu^*_{s,a} = 0$, satisfies $\mu^*(s, a)p_k(s, a) = 0$ by KKT-conditions, and $\mu^* \geq 0$ (since it is a Lagrange dual variable), thus implying that $\mu^* = \mathbf{0}$.

Intuitively, the expression in Equation 16 assigns higher probability to state-action tuples with high Bellman error $|Q_k - \mathcal{B}^*Q_{k-1}|$, but only when the *post-update* Q-value $Q_k$ is close to $Q^*$. Hence we obtain the required theorem.

**Summary of the derivation.** To summarize, our derivation for the optimal $p_k$ consists of the following key steps:

- Use the Fenchel-Young inequality to get a convenient form for the objective.

- Compute the Lagrangian, and use the implicit function theorem to compute gradients of the Q-function $Q_k$ with respect to the distribution $p_k$.

- Compute the expression for optimal $p_k$ by setting the Lagrangian gradient to $0$.

## B  Proofs for Tractable Approximations in Section 4

Here we present the proofs for the arguments behind each of the approximations described in Section 4.

**Computing weights $w_k$ for re-weighting the buffer distribution, $\mu$.** Since sampling directly from $p_k$ may not be easy, we instead choose to re-weight samples transitions drawn from a replay buffer $\mu$, using weights $w_k$ to make it as close to $p_k$. How do we obtain the exact expression for $w_k(s,a)$? One option is to apply importance sampling: choose $w_k$ as the importance ratio, $w_k(s,a) = \frac{p_k(s,a)}{\mu(s,a)}$, however this suffers from two problems – (1) importance weights tend to exhibit high variance, which can be detrimental for stochastic gradient methods; and (2) densities $\mu(s,a)$, needed to compute $w_k$ are unknown.

In order to circumvent these problems, we solve a different optimization problem, shown in Problem 17 to find the optimal *surrogate* projection distribution $q_k$, which is closest to $p_k$, under the expected likelihood metric, $\mathbb{E}_{q_k}[\log p_k]$, and at the same time closest to $\mu$ as well under the KL-divergence metric, trading off these quantities by a factor $\tau$.

$$q_k^* = \arg\min_{q_k} \mathbb{E}_{q_k}[\log p_k] + (\tau)\mathrm{D}_{\mathrm{KL}}(q_k || \mu) \tag{17}$$

where $\lambda$ is a temperature hyperparameter that trades of bias and variance. The solution to the above optimization is shown in Equation 18. The statement follows by an application of Equation 16 and manipulating the importance ratio, $\frac{q_k(s,a)}{\mu_k(s,a)}$, as the weights $w_k$.

$$q_k^*(s,a) \propto (\mu_k) \cdot \exp\left(\frac{\log p_k(s,a)}{\tau}\right)$$
$$\therefore \frac{q_k^*}{\mu_k} \propto \exp\left(\frac{-|Q_k - Q^*|(s,a)}{\tau}\right) \frac{|Q_k - \mathcal{B}^* Q_{k-1}|(s,a)}{\lambda^*} \tag{18}$$

Our next proof justifies the usage of the estimate $\Delta_k$, which is a worst-case upper bound on $|Q_k - Q^*|$ in Equation 18.

**Proof of Theorem 4.2.** We now present a Lemma B.1 which proves a recursive inequality for $|Q_k - Q^*|$, then show that the corresponding recursive estimator upper bounds $|Q_k - Q^*|$ pointwise in Lemma B.2, and then finally show that our chosen estimator $\Delta_k$ is equivalent to this recursive estimator in Theorem B.3 therefore proving Theorem 4.2.

**Lemma B.1.** *For any $k \in \mathbb{N}$, $|Q_k - Q^*|$ satisfies the following recursive inequality, pointwise for each $s, a$:*

$$|Q_k - Q^*| \le |Q_k - \mathcal{B}^* Q_{k-1}| + \gamma P^{\pi_{k-1}}|Q_{k-1} - Q^*| + \frac{2R_{max}}{1-\gamma}\max_s \mathrm{D}_{\mathrm{TV}}(\pi_{k-1}, \pi^*).$$

*Proof.* Our proof relies on a worst-case expansion of the quantity $|Q_k - Q^*|$. The proof follows the following steps. The first few steps follow common expansions/inequalities operated upon in the work on error propagation in Q-learning [35].

$$
\begin{aligned}
|Q_k - Q^*| &\stackrel{(a)}{=} |Q_k - \mathcal{B}^* Q_{k-1} + \mathcal{B}^* Q_{k-1} - Q^*| \\
&\stackrel{(b)}{\le} |Q_k - \mathcal{B}^* Q_{k-1}| + |\mathcal{B}^* Q_{k-1} - \mathcal{B}^* Q^*| \\
&\stackrel{(c)}{=} |Q_k - \mathcal{B}^* Q_{k-1}| + |R + \gamma P^{\pi_{k-1}} Q_{k-1} - R - \gamma P^{\pi^*} Q^*| \\
&\stackrel{(d)}{=} |Q_k - \mathcal{B}^* Q_{k-1}| + \gamma|P^{\pi_{k-1}} Q_{k-1} - P^{\pi_{k-1}} Q^* + P^{\pi_{k-1}} Q^* - P^{\pi^*} Q^*| \\
&\stackrel{(e)}{\le} |Q_k - \mathcal{B}^* Q_{k-1}| + \gamma P^{\pi_{k-1}}|Q_{k-1} - Q^*| + \gamma|P^{\pi_{k-1}} - P^{\pi^*}||Q^*| \\
&\stackrel{(f)}{\le} |Q_k - \mathcal{B}^* Q_{k-1}| + \gamma P^{\pi_{k-1}}|Q_{k-1} - Q^*| + \frac{2R_{max}}{1-\gamma}\max_s \mathrm{D}_{\mathrm{TV}}(\pi_{k-1}, \pi^*)
\end{aligned}
$$

where (a) follows from adding and subtracting $\mathcal{B}^* Q_{k-1}$, (b) follows from an application of triangle inequality, (c) follows from the definition of $\mathcal{B}^*$ applied to two different Q-functions, (d) follows from algebraic manipulation, (e) follows from an application of the triangle inequality, and (f) follows from bounding the maximum difference in transition matrices $|P^{\pi_{k-1}} - P^*|$ by maximum total variation divergence between policy $\pi_{k-1}$ and $\pi^*$, and bounding the maximum possible value of $Q^*$ by $\frac{R_{max}}{1-\gamma}$.
$\square$

We next show that an estimator that satisfies the recursive equality corresponding to Lemma B.1 is a pointwise upper bound on $|Q_k - Q^*|$.

**Lemma B.2.** *For any $k \in \mathbb{N}$, an vector $\Delta'_k$ satisfying*

$$\Delta'_k := |Q_k - \mathcal{B}^* Q_{k-1}| + \gamma P^{\pi_{k-1}} \Delta'_{k-1}. \tag{19}$$

*with $\alpha_k = \frac{2R_{\max}}{1-\gamma} \max_s \mathrm{D_{TV}}(\pi_k, \pi^*)$, and an initialization $\Delta'_0 := |Q_0 - Q^*|$, pointwise upper bounds $|Q_k - Q^*|$ with an offset depending on $\alpha_i$, i.e. $\Delta'_k + \sum_i \alpha_i \gamma^{k-i} \geq |Q_k - Q^*|$.*

*Proof.* Let $\Delta'_k$ be an estimator satisfying Equation 19. In order to show that $\Delta'_k + \sum_i \gamma^{k-i} \alpha_i \geq |Q_k - Q^*|$, we use the principle of mathematical induction. The base case, $k = 0$ is satisfied, since $\Delta'_0 + \alpha_0 \geq |Q_0 - Q^*|$. Now, let us assume that for a given $k = m$, $\Delta'_m + \sum_i \gamma^{m-i} \alpha_i \geq |Q_m - Q^*|$ pointwise for each $(s, a)$. Now, we need to show that a similar relation holds for $k = m + 1$, and then we can appeal to the principle of mathematical induction to complete the argument. In order to show this, we note that,

$$\Delta'_{m+1} = |Q_{m+1} - \mathcal{B}^* Q_m| + \gamma P^{\pi_m} \Delta'_m + \sum_i^{m+1} \gamma^{m+1-i} \alpha_i \tag{20}$$

$$= |Q_{m+1} - \mathcal{B}^* Q_m| + \gamma P^{\pi_m} (\Delta'_m + \sum_{i=0}^m \gamma^{m-i} \alpha_i) + \alpha_{m+1} \tag{21}$$

$$\geq |Q_{m+1} - \mathcal{B}^* Q_m| + \gamma P^{\pi_m} |Q_m - Q^*| + \alpha_m \tag{22}$$

$$\geq |Q_{m+1} - Q^*| \tag{23}$$

where (20) follows from the definition of $\Delta'_k$, (21) follows by rearranging the recursive sum containing $\alpha_i$, for $i \leq m$ alongside $\Delta_m$, (22) follows from the inductive hypothesis at $k = m$, and (23) follows from Lemma B.1.

Thus, by using the principle of mathematical induction, we have shown that $\Delta'_k + \sum_i \gamma^{k-i} \alpha_i \geq |Q_k - Q^*|$ pointwise for each $s, a$, for every $k \in \mathbb{N}$. $\qquad \square$

The final piece in this argument is to show, that the estimator $\Delta_k$ used by the DisCor algorithm (Algorithm 1), which is initialized randomly, i.e. not initialized to $\Delta_0 = |Q_0 - Q^*|$, still satisfies the property from Lemma B.2, possibly for certain $k \in \mathbb{N}$.

Therefore, we now show why: $\Delta_k + \sum_{i=1}^k \alpha_i \gamma^{k-i} \geq |Q_k - Q^*|$ point-wise for a sufficiently large $k$. We restate a slightly modified version of Theorem 4.2 for convenience.

**Theorem B.3** (Formal version of Theorem 4.2). *For a sufficiently large $k \geq k_0 = \frac{\log(1-\gamma)}{\log \gamma}$, the error estimator $\Delta_k$ pointwise satisfies:*

$$\Delta_k + \sum_{i=0}^k \gamma^i \alpha_{k-i} \geq |Q_k - Q^*|$$

*where $\alpha_i$'s are scalar constants independent of any state-action pair. (Note that Theorem 4.2 has a typo $\gamma^i$ instead of $\gamma^{k-i}$, this theorem presents the correct inequality.)*

**Proof.** *Main Idea/ Sketch:* As shown in Algorithm 1, the estimator $\Delta_k$ is initialized randomly, without taking into account $|Q_0 - Q^*|$. Therefore, in this theorem, we want to show that *irrespective* of the initialization of $Q_0$, a randomly initialized $\Delta_k$ eventually satisfies the inequality shown in Theorem 4.2. Now, we present the formal proof.

Consider $k_0$ to be the smallest $k$, such that the following inequality is satisfied:

$$\gamma^k \max_{Q_0, Q^*} |Q_0 - Q^*| \leq 1 \tag{24}$$

Thus, $k_0 \geq \frac{\log(1-\gamma)}{\log \gamma}$, assuming $R_{\max} = 1$ without loss of generality. For a different reward scaling, the bound can be scaled appropriately. To see this, we substitute $|Q_0 - Q^*|$ as an upper-bound $R_{\max}/(1-\gamma)$, and bound $R_{\max}$ by 1.

Let $\Delta'_k$ correspond to the upper-bound estimator as derived in Lemma B.2. For each $k \geq k_0$, the contribution of the initial error $|Q_0 - Q^*|$ in $|Q_k - Q^*|$ is upper-bounded by 1, and gradually decreases with a rate $\gamma$ as more backups are performed, i.e., as $k$ increases. Thus we can construct another sequence $\Delta_1, \cdot, \Delta_k, \cdots$ which chooses to ignore $|Q_0 - Q^*|$, and initializes $\Delta_0 = 0$ (or randomly) and the sequences $\Delta$ and $\Delta'_k$ satisfy:

$$|\Delta'_k - \Delta_k| < 1, \forall k \geq k_0 \tag{25}$$

Furthermore, the difference $|\Delta'_k - \Delta_k|$ steadily shrinks to 0, with a linear rate $\gamma$, so the overall contribution of the initialization sub-optimality $|Q_0 - Q^*|$ drops linearly with a rate of $\gamma$. Hence, $\Delta'$ and $\Delta$ converge to the same sequence beyond a fixed $k = k_0$. Since $\Delta'_k$ is computed using the RHS of Lemma B.1, it is guaranteed to be an upper bound on $|Q_k - Q^*|$:

$$\left| \left( \Delta_k + \sum_{i=1}^{k} \gamma^{k-i} \alpha_i \right) - \left( \Delta'_k + \sum_{i=1}^{k} \gamma^{k-i} \alpha_i \right) \right| \leq 1. \tag{26}$$

Since, $\Delta'_k + \sum_i \gamma^{k-i} \alpha_i \geq |Q_k - Q^*|$, we get $\forall\, k \geq k_0$, using 26, that

$$\Delta_k + \sum_{i=1}^{k} \gamma^{k-i} \alpha_i \geq |Q_k - Q^*| - \gamma^{k-k_0}. \tag{27}$$

Hence, $\Delta_k + \sum_{i=1}^{k} \gamma^{k-i} \alpha_i \geq |Q_k - Q^*|$ for large $k$.

*A note on the value of $k_0$.* For a discounting of $\gamma = 0.95$, we get that $k_0 \approx 59$ and for $\gamma = 0.99$, $k_0 \approx 460$. In practical instances, an RL algorithm takes a minimum of about $\geq$ 1M gradient steps, so this value of $k_0$ is easy achieved. Even in the gridworld experiments presented in Section 7.1, $\gamma = 0.95$, hence, the effects of initialization stayed significant only until about 59 iterations during training, out of a total of 300 or 500 performed, which is a small enough percentage.

**Summary of Proof for Theorem 4.2.** $\Delta_k$ in DisCor is given by the quantity $\Delta_k = |Q_k - \mathcal{B}^* Q_{k-1}| + \gamma P^{\pi_{k-1}} \Delta_{k-1}$, is an upper bound for the error $|Q_k - Q^*|$, and we can safely initialize the parametric function $\Delta_\phi$ using standard neural network initialization, since the value of initial error will matter *only* infinitesimally after a large enough $k$.

As $k \to \infty$, the following is true:

$$\lim_{k \to \infty} \left| |\Delta_k - |Q_k - Q^*|| \right| \leq \lim_{k \to \infty} \sum_{i=1}^{k} \gamma^{k-i} \alpha^i \tag{28}$$

$$= \lim_{k \to \infty} \sum_{i=1}^{k} \gamma^{k-i} \mathrm{D_{TV}}(\pi_i, \pi^*) \tag{29}$$

Also, note that if $\pi_k$ is improving, i.e. $\pi_k \to \pi^*$, then, we have that $\mathrm{D_{TV}}(\pi_k, \pi^*) \to 0$, and since limit of a sum is equal to the sum of the limit, and $\gamma < 1$, therefore, the final inequality in Equation 29 tends to 0 as $k \to \infty$.

## C  Tractable Approximations for a Practical Algorithm

We now discuss how to put together all tractable approximations discussed in Section 4 to go from the optimal distribution $p_k$ (or $w_k$) to a tractable expression for weights, that downweight the states with high estimated target value error.

We have noted all practical approximations to the expression for optimal $p_k$ (Equation 3), including estimating surrogates for $Q_k$ and $Q^*$, and the usage of importance weights to develop a method that can achieve the benefits of the optimal distribution, simply by *re-weighting transitions* in the replay buffer, *rather than altering the exploration strategy*. We also discussed a technique to reduce the variance of weights used for this reweighting. We now put these techniques together to obtain the final, practically tractable expression for the weights used for our practical approach.

We note that the term $|Q_k - Q^*|$, appearing inside the exponent in the expression for $w_k$ in Equation 6 can be approximated by the tractable upper bound $\Delta_k$. However, computing $\Delta_k$ requires the quantity

$|Q_k - \mathcal{B}^* Q_{k-1}|$ which also is unknown when $w_k$ is being chosen. Combining the upper bound on $|Q_k - \mathcal{B}^* Q_{k-1}| \leq c_2$, Theorem 4.2 and Equation 5, we obtain the following bound:

$$|Q_k - Q^*| \leq \gamma P^{\pi_{k-1}} \Delta_{k-1} + c_2 + \sum_i \gamma^i \alpha_i \tag{30}$$

Using this bound in the expression for $w_k$, along with the lower bound, $|Q_k - \mathcal{B}^* Q_{k-1}| \geq c_1$, we obtain the following lower bound on weights $w_k$:

$$w_k \propto \exp\left( \frac{-c_2 - \gamma \left[ P^{\pi_{k-1}} \Delta_{k-1} \right](s,a)}{\tau} \right) \frac{c_1}{\lambda^*} \tag{31}$$

Finally, we note that using a worst-case lower bound for $w_k$ (Equation 31) will down-weight some additional transitions which in reality lead to low error accumulation, but this scheme will never up-weight a transition with high error, thus providing for a "conservative" distribution. A less conservative expression for getting these weights is a subject of future work. Simplifying the constants $c_1$, $c_2$ and $\lambda^*$, the final expression for the practical choice of $w_k$ is:

$$w_k(s,a) \propto \exp\left( -\frac{\gamma \left[ P^{\pi_{k-1}} \Delta_{k-1} \right](s,a)}{\tau} \right). \tag{32}$$

# D    Proof From Section 3

In this section, we provide the omitted proof from Section 3 of this paper. Before going into the proofs, we first describe notation and prove some lemmas that will be useful later in the proofs.

We also describe the underlying ADP algorithm we use as an ideal algorithm for the proofs below.

---

**Algorithm 2** Generic ADP algorithm

---

1: Initialize Q-values $Q_0$.
2: **for** step $t$ in $\{1, \ldots, N\}$ **do**
3:     Collect trajectories using $\pi_t$
4:     Choose distribution $D_t$ for projection.
5:     $Q_{t+1} \leftarrow \prod_{D_t} \mathcal{B}^* Q_t$
        $\prod_{D_t} \mathcal{B}^* = \arg\min_Q \mathbb{E}_{D_t}[(Q(s,a) - \mathcal{B}^* Q_{t-1}(s,a))^2]$
6: **end for**

---

**Assumptions.**    The assumptions used in the proofs are as follows:

- Q-function is linearly represented, i.e. given a set of features, $\phi(s,a) \in \mathbb{R}^d$ for each state and action, concisely represented as the matrix $\Phi \in \mathbb{R}^{|S||A| \times d}$, Q-learning aims to learn a d-dimensional feature vector $w$, such that $Q(s,a) = w^T \phi(s,a)$. Linear function approximation is not a limiting factor in this case as we will argue in Assumption D.1, for problems with sufficiently large $|S|$ and $|A|$.

## D.1    Suboptimal Convergence of On-policy Q-learning

We first discuss a prior result from Farias and Roy [9] that describes how Q-learning can converge sub-optimally when performed with on-policy distributions, thus justifying our empirical observation of suboptimal convergence with on-policy distributions.

**Theorem D.1** ([9]). *Projected Bellman optimality operator under the on-policy distribution $\mathcal{H} = \prod_{D_\pi} \mathcal{B}^*$ with a Boltzmann policy, $\pi \propto \exp(Q/\tau)$, where $0 < \tau$ always has one or more fixed points.*

*Proof.* This statement was proven to be true in [9], where it was shown the projection operator $\mathcal{H}$ has the same fixed points as another operator, $F_\alpha$ given by:

$$F_\alpha(x) := x + \alpha \Phi^T D_\pi \left( \mathcal{B}^* \Phi x - \Phi x \right) \tag{33}$$

where $\alpha \in (0,1)$ is a constant. They showed that the operator $F_\alpha$ is a contraction for small-enough $\alpha$ and used a compact set argument to generalize it to other positive values of $\alpha$. We refer the reader to [9] for further reference.

They then showed a 2-state MDP example (Example 6.1, [9]) such that the Bellman operator $\mathcal{H}$ has **2** fixed points, thereby showing the existence of one or more fixed points for the on-policy Bellman backup operator.

This provides some theoretical evidence behind our observation in Figure 3(left), where we observed that learning to a suboptimal policy and the error plateaued, and this result provides some theoretical justification behind this empirical observation. □

### D.2 Proof of Theorem 3.1

We now provide an existence proof which highlights the difference in the speeds of learning accurate Q-values from online or on-policy and replay buffer distributions versus a scheme like DisCor. We first state an assumption (Assumption D.1) on the linear features parameterization used for the Q-function. This assumption ensures that the optimal Q-function exists in the function class (i.e. linear function class) used to model the Q-function. This assumption has also been used in a number of recent works including [7]. Analogous to [7], in our proof, we show that this assumption is indeed satisfied for features lying in a space that is logarithmic in the size of the state-space. For this theorem, we present an episodic example, and operate in a finite horizon setting with discounting $\gamma$ and $H$ denotes the horizon length. An episode terminates deterministically as soon as the run reaches a terminal node – in our case a leaf node of the tree MDP, i.e. a node at level $H - 1$ – as we will see next.

**Assumption D.1.** *There exists $\delta \geq 0$, and $w \in \mathbb{R}^d$, such that for any $(s, a) \in \mathcal{S} \times \mathcal{A}$, the optimal Q-function satisfies: $|Q^*(s, a) - w^T \phi(s, a)| \leq \delta$.*

We first prove an intermediate property of $\Phi$ satisfying the above assumption that will be crucial for the lower bound argument for on-policy distributions.

**Corollary D.1.1.** *There exists a set of features $\Phi \in \mathbb{R}^{2^H \times O(H^2/\varepsilon^2)}$ satisfying assumption D.1, such that the following holds: $||I_{2^H} - \Phi\Phi^T||_\infty \leq \epsilon$.*

*Proof.* This proof builds on the existence argument presented in [7]. Using the $\varepsilon$-rank property of the identity matrix, one can show that there exists a feature set $\Phi \in \mathbb{R}^{2^H \times O(H/\varepsilon^2)}$ such that $||I_{2^H} - \Phi\Phi^T||_\infty \leq \epsilon$. Thus, we can choose any such $\Phi$, for a sufficiently low threshold $\varepsilon$. In order to assign features $\Phi$ to a state, we can simply perform an enumeration of nodes in the tree via a standard graph search procedure such as depth first search and assign a node $(s, a)$ a feature vector $\phi(s, a)$. To begin with, let's show how we can satisfy assumption D.1 by choosing a different weight vector $w_h$ for each level $h$, such that we obtain $|Q_h(s, a) - w_h^T \phi(s, a)| \leq \epsilon$. Since for each level $h$ exactly one state satisfies $Q^*(s_j, a_j) = \gamma^{H-j+1}$, so we can just let $w_j = \gamma^{H-j+1}\phi(s_j, a_j)$ and thus we are able to satisfy Assumption D.1. This is the extent of the argument used in [7].

Now we generalize this argument to find a single $w \in \mathbb{R}^d$, unlike different weights $w_h$ for different levels $h$. In order to do this, we create a new $\Phi'$, of size $\Phi' \in \mathbb{R}^{2^H \times O(H^2/\epsilon^2)}$ (note $H^2$ versus $H$ dimensions for $\Phi'$ and $\Phi$) given any $\Phi$ satisfying the argument in the above paragraph, such that

$$\Phi'(s, a) = \left[ \underbrace{0, ..., 0}_{h \times \dim(\phi(s,a))} \quad , \quad \underbrace{\Phi(s, a)}_{\dim(\phi(s,a))} , 0, ..., 0 \right] \tag{34}$$

Essentially, we pad $\Phi$ with zeros, such that for $(s, a)$ belonging to a level $h$, $\Phi'$ is equal to $\Phi$ in the $h$−th, $\dim(\phi(s, a))$-sized block.

A choice of a single $w \in \mathbb{R}^{\dim(\Phi'(s,a))}$ for $\Phi'$ is given by simply concatenating $w_1, \cdots, w_h$ found earlier for $\Phi$.

$$w = [w_1, w_2, \cdots, w_H] \tag{35}$$

It is easy to see that $w^T \Phi'$ satisfies assumption D.1. A fact that will be used in the proof for Theorem D.2, is that this construction of $\Phi'$ also satisfies: $||I_{2^H} - \Phi'\Phi'^T||_\infty \leq \epsilon$. □

We now restate the theorem from Section 3 and provide a proof below.

Figure 10: Example element of the tree family of MDPs used to prove the lower bound in Theorem D.2. Here, the depth of the tree $H = 2$. $r(s) = 0$ implies that executing any action $a_1$ or $a_2$, a reward of 0 is obtained as state $s$. $(s^*, a^*)$ is given by state marked r(s, $a_1$) = 1.

**Theorem D.2** (Exponential lower bound for on-policy distributions). *There exists a family of MDPs parameterized by $H > 0$, with $|\mathcal{S}| = 2^H$, $|\mathcal{A}| = 2$ and a set of features satisfying Assumption D.1, such that on-policy sampling distribution, i.e. $D_k = d^{\pi_k}$, requires $\Omega\left(\gamma^{-H}\right)$ exact fixed-point iteration steps in the generic algorithm (Algorithm 2) for convergence, if at all, the algorithm converges to an $\varepsilon-$accurate Q-function.*

**Proof of Theorem D.2.** *Tree Construction.* Consider the family of tree MDPs like the one shown in Figure 10. Both the transition function $T$ and the reward function $r$ are deterministic, and there are two actions at each state: $a_1$ and $a_2$. There are $H$ level of states, thereby forming a full binary tree of depth $H$. Executing action $a_1$ transitions the state to its left child int he tree and executing action $a_2$ transitions the state to its right child. There are $2^h$ states in level $h$. Among the $2^{H-1}$ states in level $H - 1$, there is one state, $s^*$, such that action $a^*$ at this state yields a reward of $r(s^*, a^*) = 1$. For other states of the MDP, $r(s, a) = 0$. This is a typical example of a sparse reward problem, generally used for studying exploration [7], however, we re-iterate that in this case, we are primarily interested in the number of iterations needed to learn, and thereby assume that the algorithm is given infinite access to the MDP, and all transitions are observed, and the algorithm just picks a distribution $D_k$, in this case, the on-policy state-action marginal for performing backups.

*Main Argument.* Now, we are equipped with a family of the described tree MDPs and a corresponding set of features $\Phi$ which can represent an $\varepsilon-$accurate Q-function. Our aim is to show that on-policy Q-learning takes steps, exponential in the horizon for solving this task.

For any stochastic policy $\pi(a|s)$, and $\bar{p}$ defined as $\bar{p} = \min_{s \in \mathcal{S}, a \in \mathcal{A}} \pi(a|s)$, $0 < \bar{p} < 0.5$, the marginal state-action distribution satisfies:

$$d^\pi(s^*, a^*) \leq \gamma^H \cdot (1 - \bar{p})^{H+1} \tag{36}$$

Since $d^\pi$ is a discounted state-action marginal distribution, another property that it satisfies is that:

$$c \leq ||d^\pi||_2 \leq \frac{1}{1 - \gamma} 2^H \tag{37}$$

where c is a constant $c > 0$. The above is true, since, there are $2^H$ states in this MDP, and the maximum values of any entry in $d^\pi$ can be $\frac{1}{1-\gamma}$ since, $1 - \gamma$ is the least eigenvalue of $(I - \gamma P^\pi)$ for any policy $\pi$, since $||P^\pi||_2 = 1$.

Now, under an on-policy sampling scheme and a linear representation of the Q-function as assumed, the updates on the weights for each iteration of Algorithm 2 are given by ($D_{\pi_k}$ represents $\text{Diag}(d^{\pi_k})$):

$$w_{k+1} = \left(\Phi^T D_{\pi_k} \Phi\right)^{-1} \Phi^T D_{\pi_k} \left(r + \gamma P^{\pi_k} \Phi w_k\right) \tag{38}$$

Now, $||D_{\pi_k} r|| \leq \gamma^H (1 - \bar{p})^{H+1} ||\phi(s^*, a^*)||$ from the property Equation 36. Hence, the maximum 2-norm of the updated $w_{k+1}$ is given by:

$$
\begin{aligned}
||w_{k+1}||_2 &\leq ||\left(\Phi^T D_{\pi_k} \Phi\right)^{-1} \Phi^T D_{\pi_k} R||_2 + \gamma || \left(\Phi^T D_{\pi_k} \Phi\right)^{-1} \Phi^T D_{\pi_k} P^{\pi_k} \Phi w_k ||_2 \\
&\leq \frac{\gamma^H (1 - \bar{p})^{H+1}}{||D_{\pi_k}||_F \cdot (1 - \varepsilon) \cdot 2^{H-1}} + \gamma ||w_k||_2 \\
&\leq \frac{\gamma^H (1 - \bar{p})^{H+1} c}{(1 - \varepsilon) \cdot 2^{H-1}} + ||w_k||_2 \\
&= (\gamma)^H \cdot c \cdot \frac{(1)}{(1 - \varepsilon) \cdot 2^{H-1}} + ||w_k||_2.
\end{aligned}
\tag{39}
$$

where the first inequality follows by an application of the triangle inequality, the second inequality follows by using the minimum value of the Frobenius norm of the matrix $\Phi$ to be $(1 - \varepsilon) \cdot 2^{H-1}$ (using the $\varepsilon-$rank lemma used to satisfy Assumption D.1) in the denominator of the first term, bounding $||D_{\pi_k} r||$ by Equation 36, and finally bounding the second term by $\gamma ||w_k||_2$, since the maximum eigenvalue of the entire matrix in front of $w_k$ is $\leq 1$, as it is a projection matrix with a discount $\gamma$ valued scalar multiplier. The third inequality follows from lower bounding $D_{\pi_k}$ by $c$ using Equation 37.

The optimal $w^*$ is given by the fixed point of the Bellman optimality operator, and in this case satisfies the following via Cauchy-Schwartz inequality,

$$
\begin{aligned}
(I - \gamma P^*)\Phi w^* &= r \\
\implies ||\Phi||_F \cdot ||w^*||_2 &\geq ||(I - \gamma P^*)^{-1} r|| \geq \frac{1}{1 + \gamma} ||r||_2 \\
\implies (1 + \varepsilon) \cdot 2^{H-1} \cdot ||w^*||_2 &\geq \frac{1}{1 + \gamma} \\
\implies ||w^*||_2 &\geq \frac{1}{1 + \gamma} \cdot 2^{-H+1} \cdot (1 + \varepsilon)^{-1}
\end{aligned}
\tag{40}
$$

Thus, in order for $w_k$ to be equal to $w^*$, it must satisfy the above condition (Equation 40). If we choose an initialization $w_0 = \mathbf{0}$ (or a vector sufficiently close to 0), we can compute the minimum number of steps it will take for on-policy ADP to converge in this setting by using 39 and 40:

$$
\begin{aligned}
k &\geq \frac{(1 + \gamma)^{-1} \cdot 2^{-H+1} \cdot (1 + \varepsilon)^{-1}}{(\gamma)^H \cdot (1 - \bar{p})^H \cdot \frac{(c)}{(1-\varepsilon) \cdot 2^{H-1}}} \\
&\implies k \approx \Omega\left(\gamma^{-H}\right)
\end{aligned}
\tag{41}
$$

for sufficiently small $\varepsilon$. Hence, the bound follows.

*A note on the bound.* Since typically RL problems usually assume discount factors $\gamma$ close to 1, one might wonder the relevance is this bound in practice. We show via an example that this is indeed relevant. In particular, we compute the value of this bound for commonly used $\gamma, \bar{p}$ and $H$. For a discount $\gamma = 0.99$, and a minimum probability of $\bar{p} = 0.01$ (as it is common to use entropy bonuses that induce a minimum probability of taking each action), this bound is of the order of

$$
(\gamma \cdot (1 - \bar{p}))^H \approx 10^9 \quad \text{for H = 1000}
\tag{42}
$$

for commonly used horizon lengths of 1000 (example, on the gym benchmarks).

**Corollary D.2.1** (Extension to replay buffers). *There exists a family of MDPs parameterized by $H > 0$, with $|\mathcal{S}| = 2^H$, $|\mathcal{A}| = 2$ and a set of features $\Phi$ satisfying assumption D.1, such that ADP with replay buffer distribution takes $\Omega(\gamma^{-H})$ many steps of exact fixed-point iteration for convergence of ADP, if at all convergence happens to an $\epsilon-$accurate Q-function.*

**Proof of Corollary D.2.1.** For replay buffers, we can prove a similar statement as previously. The steps in this proof follow exactly the steps in the proof for the previous theorem.

With replay buffers, the distribution for the projection at iteration $k$ is given by:

$$
d_k(s, a) = \frac{1}{k} \sum_{i=1}^{k} d_{\pi_k}(s, a)
\tag{43}
$$

Therefore, we can bound the probability of observing any state-action pair similar to Equation 36 as:

$$d_k(s^*, a^*) \leq \frac{1}{k} \sum_{i=1}^{k} \gamma^H \cdot (1 - \bar{p})^{H+1} \qquad (44)$$

with $\bar{p}$ as defined previously. Note that this inequality is the same as the previous proof, and doesn't change. We next bound the 2-norm of the state-visitation distribution, in this case, the state-distribution in the buffer.

$$c \leq ||d_k||_2 \leq \frac{1}{1 - \gamma} \cdot 2^H \qquad (45)$$

where $c > 0$. The two main inequalities used are thus the same as the previous proof. Now, we can simply follow the previous proof to prove the result.

**Practical Implications.** In this example, both on-policy and replay buffer Q-learning suffer from the problem of exponentially many samples need to reach the optimal Q-function. Even in our experiments in Section 3, we find that on-policy distributions tend to reduce errors very slowly, at a rate that is very small. The above bound extends this result to replay buffers as well.

In our next result, however, we show that an optimal choice of distribution, including DisCor, can avoid the large iteration complexity in this family of MDPs. Specifically, using the errors against $Q^*$, i.e. $|Q_k - Q^*|$ can help provide a signal to improve the Q-function such that this optimal distribution / DisCor will take only $\text{poly}(H)$ many iterations for convergence.

**Theorem D.3** (Optimal distributions / DisCor). *In the tree MDP family considered in Theorem 3.1, with linear function approximation for the Q-function, and with Assumption D.1 for the features $\Phi$, DisCor takes $\text{poly}(H)$ many exact iterations for $\varepsilon-$accurate convergence to the optimal Q-function.*

*Proof.* We finally show that the DisCor algorithm, which prioritizes states based on the error in target values, will take $\text{poly}(H)$ many steps for convergence. Assume that Q-values are initialized randomly, for example via a normal random variable with standard deviation $\sigma$, i.e., $Q_0(s, a) \sim \mathcal{N}(0, \sigma^2)$, however, $\sigma$ is very small, but is more than 0 ($\sigma > 0$) (this proof is still comparable to the proof for on-policy distributions, since Q-values can also be initialized very close to 0 even in that case, and the proof of Theorem D.2 still remains valid.).

Now we reason about a run of DisCor in this case.

**Iteration 1.** In the first iteration, among all nodes in the MDP, the leaf nodes (depth $H$-1) have 0 error at the corresponding target values, since an episode terminates once a rollout reaches a leaf node. Hence, the algorithm will assign equal mass to all leaf node states, and exactly update the Q-values for nodes in this level (upto $\varepsilon$-accuracy).

**Iteration 2.** In the second iteration, the leaf nodes at level $H - 1$ have accurate Q-values, therefore, the algorithm will pick nodes at the level $H - 2$, for which the target values, i.e. Q-values for nodes at level $H - 1$, have 0 error. The algorithm will update Q-values at these nodes at level $H - 2$, while ensuring that the incurred error at the nodes at level $H - 1$ isn't beyond $\varepsilon$. Since, the optimal value function $Q^*$ can be represented upto $\varepsilon-$accuracy, we can satisfy this criterion.

**Iteration $k$.** In iteration $k$, the algorithm updates Q-values for nodes at level $H - k$, while also ensuring Q-values for all nodes at a level higher than $H - k$ are estimated within the range of $\varepsilon-$allowable error. This is feasible since, $Q^*$ is expressible with $\varepsilon-$accuracy within the linear function class chosen.

This iteration process continues, and progress level by level, from the leaves (level $H - 1$) to the root (level 0). At each iteration Q-values for all states at the same level, and below are learned together. Since learning progresses in a "one level at-a-time" fashion, with guaranteed correct target values (i.e. target values are equal to the optimal Q-function $Q^*$) for any update that the algorithm performs, it would take at most $\text{poly}(H)$ many iterations (for example, multiple passes through the depth of the tree) for $\varepsilon$-accurate convergence to the optimal $Q$-function. □

# E   Extended Related Work

In this section, we discuss, in detail, the relationships between prior works, discussed in Section 6 and our work, including the proposed algorithm, DisCor.

**Error propagation in ADP.** A number of prior works have analysed error propagation in ADP methods. Most work in this area has been devoted to analysing how errors in Bellman error minimization propagate through the learning process of the ADP algorithm, typically focusing on methods such as fitted Q-iteration (FQI) [38] or approximate policy iteration [37]. Prior works in this area assume an abstract error model, and analyze how errors propagate. Typically these prior works only limitedly explore reasons for error propagation or present methods to curb error propagation. [34] analyze error propagation in approximate policy iteration methods using quadratic norms. [35] analyze the propagation of error across iterations of approximate value iteration (AVI) for $L_p$-norm $p = (1, 2)$. [36] provide finite sample guarantees of AVI using error propagation analysis. Similar ideas have been used to provide error bounds for a number of different methods – [8, 43, 26, 42] and many more. In this work, we show that ADP algorithms suffer from an absence of corrective feedback, which arises because the data distribution collected by an agent is insufficient to ensure that error propagation is eventually corrected for. We further propose an approach, DisCor, which can be used in conjunction with modern deep RL methods.

**Offline / Batch Reinforcement Learning.** Our work bears some similarity to the recent body of literature on batch, or offline reinforcement learning [25, 12, 55], where the goal is to learn an effective policy, using access to only a finite, off-policy dataset collected previously. All of these works augment ADP methods with additional constraints on the policy to be close to the data-collection policy, under some closeness metric. While [25] show that this choice can be motivated from the perspective of error propagation, we note that there are clear differences between our work and such prior works in offline RL. First, the problem statement of offline RL requires learning from completely offline experience, however, our method learns online, via on-policy interaction and a replay buffer. While error propagation due to unobserved state-action pairs [25, 12] is the primary problem behind incorrect Q-functions in offline RL, in this paper, firstly, we show that such error accumulation also happens in online reinforcement learning, which results in a lack of corrective feedback, and secondly, the primary reason behind such error propagation is an interaction between data distribution and function approximation.

**Generalization effects in deep Q-learning.** There are a number of recent works that theoretically analyze and empirically demonstrate that certain design decisions for neural net architectures used for Q-learning, or ADP objectives can prove to be significant in deep Q-learning. For instance, [30] point out that sparse representations may help Q-learning algorithms, which links back to prior literature on state-aliasing and destructive interference. [1] uses an objective inspired from the neural tangent kernel (NTK) [21] to "cancel" generalization effects in the Q-function induced across state-action pairs to mimic tabular and online Q-learning. Our approach, DisCor, can be interpreted as *only* indirectly affecting generalization via the target Q-values for state-action pairs that will be used as bootstrap targets for the Bellman backup, which are expected to be accurate with DisCor, and this can aid generalization, similar to how generalization can be achieved via abstention from training on noisy labels in supervised learning [48].

# F  Experimental and Implementation Details

In this section, we provide experimental details, such as the DisCor algorithm in practice (Section F.1), and the hyperparameter choices (Section F.2).

## F.1  DisCor in Practice

In this section, we provide details on the experimental setup and present the pseudo-code for the practical instantiation of our algorithm, DisCor.

The pseudocode for the practical algorithm is provided in Algorithm 3. Like any other ADP algorithm, such as DQN or SAC, our algorithm maintains a pair of Q-functions – the online Q-network $Q_\theta$ and a target network $Q_{\bar\theta}$. For continuous control domains, we use the clipped double Q-learning trick [11], which is also referred to as the "twin-Q" trick, and it further parametrizes another pair of online and target Q-functions, and uses the minimum Q-value for backup computation. In addition to Q-functions, in a continuous control domain, we parametrize a separate policy network $\pi_\psi$ similar to SAC. In a discrete action domain, the policy is just given by a greedy maximization of the online Q-network.

**Algorithm 3 DisCor: Deep RL Version**

1: Initialize online Q-network $Q_\theta(s, a)$, target Q-network, $Q_{\bar\theta}(s, a)$, error network $\Delta_\phi(s, a)$, target error network $\Delta_{\bar\phi}$, initial distribution $p_0(s, a)$, a replay buffer $\beta$ and a policy $\pi_\psi(a|s)$, number of gradient steps $G$, target network update rate $\eta$, initial temperature for computing weights $w_k$, $\tau_0$.
2: **for** step $k$ in $\{1, \dots, \}$ **do**
3:     Collect $M$ samples using $\pi_\psi(a|s)$, add them to replay buffer $\beta$, sample $\{(s_i, a_i)\}_{i=1}^N \sim \beta$
4:     Evaluate $Q_\theta(s, a)$ and $\Delta_\phi(s, a)$ on samples $(s_i, a_i)$.
5:     Compute target values for $Q$ and $\Delta$ on samples:

$$y_i = r_i + \gamma \mathbb{E}_{a' \sim \pi_\psi(a'|s')}[Q_{\bar\theta}(s_i', a')]$$

$$\hat\Delta_i = |Q_\theta(s, a) - y_i| + \gamma \mathbb{E}_{\hat a_i \sim \pi(a_i|s')}[\Delta_{\bar\phi}(s_i', \hat a_i)]$$

6:     Compute $w_k$ using Equation 7 with temperature $\tau_k$
7:     Take $G$ gradient steps on the Bellman error for training $Q_\theta$ weighted by $w_k$.

$$\theta \leftarrow \theta - \alpha \nabla_\theta \frac{1}{N} \sum_{i=1}^N w_k(s_i, a_i) \cdot (Q_\theta(s_i, a_i) - y_i)^2$$

8:     Tale $G$ gradient steps to minimize unweighted (regular) Bellman error for training $\phi$.

$$\phi \leftarrow \phi - \alpha \nabla_\phi \frac{1}{N} \sum_{i=1}^N (\Delta_\theta(s_i, a_i) - \hat\Delta_i)^2$$

9:     Update the policy $\pi_\psi$ if it is explicitly modeled.

$$\psi \leftarrow \psi + \alpha \nabla_\psi \mathbb{E}_{s \sim \beta, a \sim \pi_\psi(a|s)}[Q_\theta(s, a)]$$

10:    Update target networks using soft updates (SAC), hard updates (DQN)

$$\bar\theta \leftarrow (1 - \eta)\bar\theta + \eta\theta$$

$$\bar\phi \leftarrow (1 - \eta)\bar\phi + \eta\phi$$

11:    Update temperature hyperparameter for DisCor:

$$\tau_{k+1} \leftarrow (1 - \eta)\tau_k + \eta \ \text{BATCH-MEAN}(\Delta_\phi(s_i, a_i))$$

12: **end for**

---

DisCor further maintains a model for accumulating errors $\Delta_\phi$ parameterized by $\phi$ and the corresponding target error network $\Delta_{\bar\phi}$. In the setting with two Q-functions, DisCor models two networks, one for modelling error in each Q-function. At every step, a few (depending upon the algorithm) gradient steps are performed on $Q$ and $\Delta$, and $\pi$ – if it is explicitly modeled, for instance in continuous control domains. This is a modification of generalized ADP Algorithm 2 and the corresponding DisCor version (Algorithm 1), customized to modern deep RL methods.

### F.2 Experimental Hyperparameter Choices

We finally specify the hyperparameters we used for our experiments. These are as follows:

- *Temperature $\tau$:* DisCor mainly introduces one hyperparameter, the temperature $\tau$ used to compute the weights $w_k$ in Equation 7. As shown in Line 11 of Algorithm 3, DisCor maintains a moving average of the temperatures and uses this average to perform the weighting. This removes the requirement for tuning the temperature values at all. For initialization, we chose $\tau_0 = 10.0$ for all our experiments, irrespective of the domain or task.

  In our preliminary experiments, we tried experimenting with a fixed temperature, which did not yield good results, and suffered from either too large importance weights, giving rise to high variance in the learning curve or wasn't effective beyond SAC at all.

- *Architecture for $\Delta_\phi$:* For the design of the error network, $\Delta_\phi$, we utilize a network with 1 extra hidden layer than the corresponding Q-network. For instance, in metaworld domains, the standard Q-network used was [256, 256, 256] in size, and thus we used an error network of size: [256, 256, 256, 256], and for MT10 tasks we used [160, 160, 160, 160, 160, 160] sized Q-networks [57] and 1-extra layer error networks $\Delta_\phi$.

- *Target net updates:* We performed target net updates for $\Delta_{\bar{\phi}}$ in the same manner as standard Q-functions, in all domains. For instance, in MetaWorld, we update the target network $\Delta_{\bar{\phi}}$ with a soft update rate of 0.005 at each environment step, as is standard with SAC [51], whereas in DQN [33], we use hard target resets.

- *Learning rates for $\Delta_{\phi}$:* These were chosen to be the same as the corresponding learning rate for the Q-function, which is $3e-4$ for SAC and $0.0025$ for DQN. We also searched over the space of three learning rates: $[3e-4, 1e-4, 5e-4]$, and did not find a huge difference across these. The only somewhat visible difference suggested that a learning rate of $3e-4$ or $5e-4$ worked better than 1e-4..

- *Official Implementation repositories used for our work:*

    1. Soft-Actor-Critic [14]: `https://github.com/rail-berkeley/softlearning/`
    2. Dopamine [4]: Offical DQN implementation `https://github.com/google/dopamine`, and the baseline DQN numbers were reported from the logs available at: `https://github.com/google/dopamine/tree/master/baselines`
    3. Gridworlds [10]: `https://github.com/justinjfu/diagnosing_qlearning`

- We perform self-normalized importance sampling across a batch, instead of regular importance sampling, since that gives rise to more stable training, and suffers less from the curse of variance in importance sampling.

- *Seeds*: In all our experiments, we implemented our methods on top of the official repositories, ran each experiment for 4 randomly chosen seeds from the interval, $[10, 10000]$, in Meta-World, OpenAI gym and tabular environments. For DQNs on atari, we were only able to run 3 seeds for each game for our method, however, we found similar performances, and less variance across seeds, as is evident from the variance bands in the corresponding results. For baseline DQN, we just used the log files provided by the dopamine repository for our results.

## G  Additional Experiments

We now present some additional experimental results which we could not present de to shortage of space in Section 7.

### G.1  Tabular Environment Analysis

**Environment Setup.** We used the suite of tabular environments from from Fu et al. [10], which provides a suite of 8 tabular environments and several different plug-and-play options for choices of input feature space, reward functions, etc. a suite of algorithms based on fitted Q-iteration [38], which forms the basis of modern deep RL algorithms that use ADP. We evaluated performance on different variants of the $(16, 16)$ gridworld provided, with different reward styles (sparse, dense), different observation functions (one-hot, random features, locally smooth observations), and different amounts of entropy coefficients (0.01, 0.1). We evaluated on five different kinds of environments: grid16randomobs, grid16onehot, grid16smoothobs, grid16smoothsparse, grid16randomsparse – which cover a wide variety of combinations of feature and reward types. We also evaluated on CliffWalk, Sparsegraph and MountainCar MDPs in Figures 11 and 12.

**Sampling Modes.** We evaluated in two modes – (1) exact mode, in the absence of sampling error, where an algorithm is provided with all transitions in the MDP and simply chooses a weighting over the states rather than sampling transitions from the environment, and (2) sampled mode, which is the conventional RL paradigm, where the algorithm performs online data collection to collect its own data.

**Setup for Figure 3.** For Figure 3, we used the grid16randomobs MDP (which is a $16 \times 16$ gridworld with randomly initialized vectors as observations), with an entropy penalty of 0.01 to the policy. For Figure 3 (left) we used the grid16smoothobs MDP with locally smooth observations, with an entropy penalty of 0.01 as well, and for Figure 3 (right), we used grid16smoothsparse environment, with sparse reward and smooth features.

Figure 11: Performance of different methods: DisCor (blue), DisCor (oracle) (red), Replay buffer Q-learning (green, on-policy (grey) and prioritized updates (orange), across different environments measured in terms of smooth normalized returns in the **exact** setting with all transitions. Note that DisCor and DisCor (oracle) generally tend to perform better.

Figure 12: Performance of different methods: DisCor (blue), DisCor (oracle) (red), Replay buffer Q-learning (green) and prioritized updates (orange). across different environments measured in terms of smooth normalized return with **sampled** transitions. Note that DisCor and DisCor (oracle) generally tend to perform better.

**Results.** We provide some individual environment performance curves showing the smoothed normalized return achieved at the end of 300 steps of training in both exact (Figure 11) and sampled (Figure 12) settings. We also present some individual-environment learning curves for these environments comparing different methods in both exact (Figure 13) and sampled (Figure 14).

### G.2 MetaWorld Tasks

In this section, we first provide a pictorial description of the six hard tasks we tested on from metaworld, where SAC usually does not perform very well. Figure 15 shows these tasks. We provide the trends for average return achieved during evaluation (not the success rate as shown in Figure 5 in Section 7) for each of the six tasks. Note that DisCor clearly outperforms both the baseline SAC and the prior method PER in all six cases, achieving nearly **50%** more than the returns achieved by SAC.

### G.3 OpenAI Gym Benchmarks

Here we present an evaluation on the standard OpenAI continuous control gym benchmark environments. Modern ADP algorithms such as SAC can already solve these tasks easily, without any issues, since these algorithms have been tuned on these tasks. A comparison of the three algorithms DisCor, SAC and PER, on three of these benchmark tasks is shown in Figure 17 (top). We note that in this case, all the algorithms are roughly comparable to each other. For instance, DisCor performs better than SAC and PER on Walker2d, however, is outperformed by SAC on Ant.

Figure 13: Learning curves for different algorithms in the **exact** setting. Note that DisCor (blue) and DisCor (oracle) (red) are generally the best algorithms in these settings. Replay Buffers (green) help over on-policy (pink) distributions. Prioritizing transitions based on high Bellman error (orange) is performant in some cases, but hurts in the other cases – it is especially slow in cases with sparse rewards, note the speed of learning on grid16randomsparse and grid16smoothsparse (**right** of the vertical line) environments.

Figure 14: Learning curves for different algorithms in the **sampled** setting. Note that DisCor and DisCor (oralce) anre generally the best algorithms in these settings. Replay Buffers (green) help over on-policy (gray) distributions, but may the algorithm may still fail to reach optimal return. Prioritizing for high Bellman error (PER) may fail to learn in sparse-reward tasks as is evident from the curves for sparse reward environments (**right** of the vertical line).

hammer · push with stick · push with wall

pull with stick · turn dial · insert peg side

Figure 15: Visual description of the six MetaWorld tasks used in our experiments in Section 7. Figures taken from [56].

**Stochastic reward signals.** That said, we also performed an experiment to verify the impact of stochasticity, such as noise in the reward signal, on the DisCor algorithm as compared to other baseline algorithms like SAC and PER. Analogous the diagnostic tabular experiments on low signal-to-noise ratio environments, such as those with sparse reward, we would expect a baseline ADP method to be impacted more due to an absence of corrective feedback in tasks with stochastic reward noise,

Figure 16: Evaluation average return achieved by DisCor (blue), SAC (green) and PER (orange) on six Metaworld benchmarks. From left to right: pull stick, push with wall, push with stick, turn dial, hammer and insert peg side tasks. Note that DisCor clearly achieves better returns or learns faster in most of the tasks.

since a noisy reward effectively reduces the signal-to-noise ratio. We would also expect a method that ensures corrective feedback to perform better.

In order to test this hypothesis, we created stochastic reward tasks out of the OpenAI gym benchmarks. We modified the reward function $r(s, a)$ in these gym tasks to be equal to:

$$r'(s, a) = r(s, a) + z, \quad z \sim \mathcal{N}(0, 1) \tag{46}$$

and the agent is only provided these noisy rewards during training. However, we only report the deterministic ground-truth reward during evaluation. We present the results in Figure 17 (bottom). Observe that in this scenario, DisCor emerges as the best performing algorithm on these tasks, and outperforms other baselines SAC and PER both in terms of asymptotic performance (example, HalfCheetah) and sample efficiency (example, Ant).

We also compare DisCor to AFM [10], a prior method similar to prioritized experience replay on the MuJoCo gym benchmarks. We find that DisCor clearly outperforms AFM in these scenarios. We present these results in Figure 17 (bottom) where the top row presents results in the case of regular gym benchmarks, and the bottom row presents results in the case of gym benchmarks with stochastic reward noise.

Figure 17: Performance of DisCor, SAC, PER and AFM on continuous control gym benchmarks (top row) and gym benchmarks *with stochastic reward noise* (bottom row). Observe that DisCor learns slightly faster and performs better than SAC and PER on these stochastic problems and that DisCor clearly out-performs AFM in both scenarios, on all three benchmarks tested on.

## G.4 MT10 Multi-Task Experiments

In this section, we present the trend of returns, as a learning curve and as a comparative histogram (at 1M environment steps of training) for the multi-task MT10 benchmark, extending the results shown in Section 7.3, Figure 8. These plots are shown in Figure 18. Observe that DisCor achieves more than **30%** of the return of SAC, and obtains an individually higher value of return on more tasks.

(a) Average task return

(b) Per-task return at 1M steps

Figure 18: Performance of DisCor and SAC on the MT10 benchmark. Returns for DisCor are higher than SAC by around **30%**; (2) DisCor achieves a non-trivial return on **7/10** tasks after 1000k steps, as compared to **3/10** for unweighted SAC, similar to the trend at 500k steps shown in Figure 8.

## G.5 MT50 Multi-Task Experiments

We further evaluated the performance of DisCor on the multi-task MT50 benchmark [56]. This is an extremely challenging benchmark where the task is to learn a single policy that can solve 50 tasks together, with the same evaluation protocol as previously used in the MT10 experiments (Section 7.3 and Appendix G.4). We present the results (average task return and average success rate) in Figures 19. Note that while SAC tends to saturate/plateau in between 4M - 8M steps, accounting for corrective feedback via the DisCor algorithm makes the algorithm continue learning in that scenario too.

(a) Average task return

(b) Average success rate

Figure 19: Performance of DisCor and SAC on the MT50 benchmark. Note that, DisCor clearly keeps learning unlike SAC which tends to plateau for about 3M steps in the middle (the stretch between 4M and 7M steps on the x-axis, where SAC exhibits a small gradient in the learning progress, whereas DisCor continuously keeps learning).

## G.6 DQN with multi-step returns

N-step returns with DQN are hypothesized to stabilize learning since updates to the Q-function now depends on reward values spanning multiple steps, and the coefficient of the bootstrapped Q-value is $\gamma^T$, which is exponentially smaller than $\gamma$ used conventionally in Bellman backups, implying that the error accumulation process due to incorrect targets is reduced. Thus, we perform a comparison of DisCor and DQN with n-step backups, where $n$ was chosen to be 3, $n = 3$, in accordance with commonly used multi-step return settings for Atari games. We present the average return obtained by DisCor and DQN (+n-step), with sticky actions, in Table 1. We clearly observe that DisCor outperforms DQN with 3-step returns in all three games evaluated on. We also observe that n-step returns applied with DisCor also outperform n-step returns applied with DQN, indicating the benefits of using DisCor even when other techniques, such as n-step returns are used.

## G.7 Code for the Method

The code is shown in Figure 20. It is a simplified version of the code from our implementation of DisCor on top of the official SAC repository [51].

| Game | n-step DQN $(n=3)$ | DisCor (Regular) | n-step DisCor $(n=3)$ |
|---|---|---|---|
| **Pong** | 17 | 17 | **19** |
| **Breakout** | 37 | **175** | 47 |

Table 1: Average Performance of DQN + 3-step returns, DisCor and Discor + 3-step returns on Pong and Breakout at 60M steps into training, rounded off to the nearest integer. Note that DisCor clearly outperforms DQN with multi-step returns. We also find that adding n-step returns to DisCor can hurt, for instance, on Breakout, where the same hurts with DQN as well (for comparison, see Figure 9 in the main paper), however, we still observe that DisCor, when applied with multi-step returns performs better than DQN with multi-step returns as well, indicating the benefits of DisCor even when methods such as multi-step returns are used.

```python
def _init_critic_update_with_dist(self):
    """Update critic with distribution weighting,
        and update \delta_\phi using recursive update. """
    next_actions = self._policy.actions([self._next_observations_ph])

    ## Compute errors at next state, and an action from the policy
    qf_pred_errs = self._error_fns([self._next_observations_ph,
next_actions])

    ## error_model_tau_ph: moving mean of the error values over
batches
    err_logits = -tf.stop_gradient(
        self._discount * qf_pred_errs / self._error_model_tau_ph)

    Q_target = tf.stop_gradient(self._get_Q_target())
    Q_values = self._Q([self._observations_ph, self._actions_ph])

    ## Compute importance sampled loss, also perform self-normalized
sampling
    loss, weights = importance_sampled_loss(
        labels=Q_target, predictions=Q_values,
        weights=err_logits, weight_options='self_normalized')

    ## Train Q-function
    Q_training_ops = tf.contrib.layers.optimize_loss(loss,
learning_rate=self._Q_lr,
        optimizer=self._Q_optimizer, variables=self._Q.
trainable_variables)
    training_ops.update({'Q': tf.group(Q_training_ops)})

    ## Training the error function
    err_values = self._error_fns([self._observations_ph, self.
_actions_ph])

    ## Mean Bellman error used to compute target values for error
    bellman_errors = tf.abs(Q_values - Q_target)
    err_targets = tf.stop_gradient(self._get_error_target(
bellman_errors))

    ## This is used to update the moving mean, self.
_error_model_tau_ph
    self._mean_error_values = tf.reduce_mean(err_values)

    ## Simple mean squared error loss for \delta_\phi
    err_losses =  tf.losses.mean_squared_error(
        labels=err_targets, predictions=err_values, weights=0.5)

    ## Update error function: \delta_\phi
    err_training_ops = tf.contrib.layers.optimize_loss(err_losses,
        learning_rate=self._dist_lr,
        optimizer=self._err_optimizer, variables=self._error_fns.
trainable_variables)
    training_ops.update({'Error': tf.group(err_training_ops)})
```

Figure 20: Code for training the error function $\Delta_\phi$, and modified training for the Q-function $Q(s, a)$ using $\Delta_\phi$ to get weights $w(s, a)$ for training. Code written in convention with regular Tensorflow guidelines, in the same style as the official SAC implementation [51].