[Reviews · NeurIPS 2020]

Review 1

Summary and Contributions: This paper proposes a modification that can be made to approximate dynamic programming (ADP) algorithms which corrects error in the value function incurred during the learning process. A thorough analysis is conducted that explores when and why these errors arise in deep RL---in short, when the bootstrapping targets in updates used by ADPs are erroneous, then using these as update targets may result in poor performance. Intuition for the proposed method is derived and supported theoretically and supplemented with examples. A strong empirical evaluation is conducted over a range of RL benchmarks, including multi-task RL, robotics, and Atari games. Key takeaways include 1) the guiding principles behind DisCor when applied in approximation (and exactly) improve performance, and 2) DisCor can learn faster on many tasks, especially those not extensively tuned and that have challenging properties (e.g., multi-task learning).

Strengths: The work provides an insightful explanation and analysis of the potential for error in ADP methods that use bootstrapped target values. The analysis provides much better understanding of this issue to the ML community. The algorithm itself is well-explained, i.e., reasoning for the method is thoroughly derived and discussed in the paper. A empirical evaluation shows very promising results over different types of RL domains, including multi-task learning, domains with stochastic discrete-actions and image-states, and continuous control tasks.

Weaknesses: I found few shortcomings in the work. The bound for the $w_k$ approximation discussed in the appendix seems a bit loose (this is already acknowledged by the authors), but this does not seem to take away from the performance in the empirical analysis.

Correctness: The empirical methodology (presented at a high level in the main body and more granularly in the appendix) seem correct. My examination of the proofs was by no means thorough, but I did not find errors following along.

Clarity: Yes! The paper was very clear, with simple language and helpful examples to supplement description of important concepts. Proofs were straightforward and supplemented with explanation that made it easier to follow.

Relation to Prior Work: Yes.

Reproducibility: Yes

Additional Feedback: line 3 "this remain unclear" -> "this remains unclear" line 43 "are themselves are erroneous" -> "are erroneous" line 92 "consider tree-structured" -> "consider the tree-structured" It might be helpful for readers who are red-green colorblind or for those printing in black/white that the colors of the graphs in Figure 1 are changed. The appendix in general was very readable! Awesome. ======POST REBUTTAL===== I have read the rebuttal. My score remains the same.


Review 2

Summary and Contributions: The authors describe a problem that occurs when using neural network function Q-function approximations when the Q-function is updated by fitting a Q-function that minimizes the empirical squared loss over the data collected previously (mixture distribution over past policies). They describe the optimal solution to the problem (if one had full information). Then they develop an algorithm to approximate the optimal solution to the problem (an alternative weighting over the past examples). ------------------------------------------------------------------------------------------ 8/21/2020: I feel that the authors have addressed my main concerns in their rebuttal, so I have updated my score.

Strengths: - They demonstrate strong empirical performance of their algorithm. - They illustrate the problem (that their algorithm is designed to fix) pretty well empirically (see later comments for areas to improve)

Weaknesses: - I am not fully convinced / confused about your argument regarding Figure 3. What does computing Q using Unif(s,a) instead of the on-policy distribution? Do you reweight the samples from the on-policy distribution with some kind of importance weights to achieve Unif(s,a)? Or is the Q computed using Unif(s,a) an oracle quantity that is not achievable in practice? It's not clear to me what is the exact source of the problem when using neural network function approximation---I would like to see you clarify and make this argument more precise. For simplicity, let's assume we just have Q_0 and Q_1 so initial policy and one update, and we are trying to fit Q_2. Are you arguing that since the data collected under Q_0 was used to fit Q_1, and Q_2 is fit using the data collected under Q_0 and Q_1, there is some correlation induced regarding which states are visited / values of rewards that that causes poor learning? Or is the issue that you are not visiting some state-action tuples frequently enough (far from the uniform distribution)?

Correctness: The methods and empirical methodology appear to be correct.

Clarity: I think the paper is well written overall---I urge the authors to be more precise with their claims / argument regarding the problem their algorithm is designed to fixed.

Relation to Prior Work: Yes.

Reproducibility: Yes

Additional Feedback: - I found figure 3 very difficult to interpret. I think this experiment / figure is crucial to your argument that the problem is due to a combination of (1) using neural network function approximation and (2) updating Q with the minimizer of the empirical square loss on previous trajectories. - I would like to see an intuitive explanation / discussion of the result of Theorem 4.1. How exactly is the optimal weighting different from the mixture of past policies / default \mu? What does it upweight / down-weight? - I didn't understand the color changing in Figure 1. Is this a hypothetical sequence of updates? Could this figure correspond to a more clear scenario one might encounter? - I wonder if the "correlation" problem you're describing is related to the problem of inference on bandit data e.g. like in https://arxiv.org/abs/1911.02768 and https://arxiv.org/abs/1708.01977, that estimators like the sample mean can be biased and asymptotically non-normal on bandit data due to correlation between previous reward and the action selection policy. - Is there literature on the correspondence between optimizing (2) and minimizing regret that you could reference? Since your empirical experiments demonstrate improvements in regret, while your algorithm optimizes to minimize error in the Q function.


Review 3

Summary and Contributions: The authors suggest a concern in instability in Q-learning, a lack of corrective feedback in RL, where increased visitation to a state-action pair doesn't result in lower error to Q*. A method is introduced which reweights samples to approximate the optimal distribution such that the distance between Q and Q* is minimized at a given time step. Experimental validation is performed on MetaWorld tasks where the proposed method outperforms SAC.

Strengths: The proposed method is fairly simple and produces a meaningful gain in improvement on the MetaWorld tasks. The idea to the approach is also quite unique/thought-provoking. The supplementary material is very thorough with additional details and experiments and code is provided. The method feels reproducible.

Weaknesses: The authors described a lack of corrective feedback as an issue with function approximation + RL, however it was unclear to me why this problem was unique to DRL. For example, even in tabular Q-learning with stochastic transitions or rewards, it is easy to imagine a scenario where the distance of Q(s,a) to Q*(s,a) can grow with additional updates. Consequently, is this truly a problem or simply a property of Q-learning? I would like to see some discussion of the tabular setting. "Value error" seems like a poor name (line 109), as under this definition the exact value of a fixed policy (Qpi) would have "value error". Suggestion: Optimality error.

Correctness: I have no issues with the correctness. Although, as the method relies on a series of approximations, the paper could be improved with discussion on the impact of these approximations.

Clarity: The paper is clearly written for the most part. I find Figure 3 (right) unclear, as the title "Sparse rewards" is unmentioned anywhere. The algorithmic description could be improved as well. Eqn (7) includes a delta term, which itself is approximated by another term (line 194).

Relation to Prior Work: To the best of my knowledge the paper is quite novel and is well-positioned. Additional related work is included in the supplementary material.

Reproducibility: Yes

Additional Feedback: Can the results be generalized to policy evaluation? --- Update --- Thank you for taking the time to respond. This is a solid paper and I advocate for acceptance, but I will not be otherwise adjusting my score.


Review 4

Summary and Contributions: This paper presents a new theoretical and practical contribution for approximate dynamic programming (ADP) algorithms such as Q-learning, actor critic, and their derivatives, DisCor. The authors first present a possible reason for the instability commonly observed in these algorithms. Whereas supervised learning tasks can benefit from corrective feedback to fix poor value estimates, the bootstrapped targets common in ADP can actually reinforce poor value estimates and result in slow or no convergence. DisCor is a modification to ADP algorithms that maintains an estimate of the true data distribution, using this to weight the effect of transitions on the main Q-model. These weightings correct for poor estimates by down-weighting those states and preferentially learning on transitions that will reduce the value error rather than the bellman error.

Strengths: Overall a powerful contribution to the field. The paper is very theoretically-grounded, with plenty of explanation of intuition and proof of the approximations used. Yet it's also empirically quite powerful, showing a significant improvement over state-of-the-art in common benchmark tasks. The significance of the contribution is large. Most RL algorithms are exactly the ADP family that this proposes to modify, and the addition of this corrective feedback model can be slotted into most training loops without compatibility issues. As the authors note, it could also be used to guide exploration rather than just for post hoc transition correction. This is clearly relevant to the NeurIPS community, much of which makes use of this form of RL algorithm. Novelty is also present. I found the reformulation in terms of value error instead of Bellman error quite powerful, and I would hope to see more research on this. Previous work in this area seems sparse; tons of research uses function approximation, but doesn't explicitly consider how the data distribution can be impacted by coupling effects. This seems a more powerful form of prioritized experience replay. The empirical evaluations are also well-done. SAC is an appropriate state-of-the-art baseline for continuous control, and the selection of tasks ranges includes tabular (used to demonstrate value error vs bellman error), continuous control, pixel-based, and even multi-task environments which aren't well-solved yet. In most cases, DisCor significantly outperforms the baseline. The comparison to an oracle DisCor also demonstrated the empirical effectiveness of the approximations used.

Weaknesses: This is likely due to space limitations, given the amount that needed to be devoted to preliminaries and reformulation of the problem, but the discussion sections are a bit sparse. It might have been helpful to see a specific example of the learning dynamics, such as comparing a particularly poor state in the baseline updates with the revised version in DisCor (similar to the tree MDP). I was also curious to know why DisCor exhibited such high variance in some of the MetaWorld tasks. I'm also curious to know the empirical impact of the extra computation done to maintain the model.

Correctness: The claims are correct, and the method and its preliminaries are explained well and derived in detail. The empirical methodology is also correct, drawing from previous baselines and benchmark tasks, building DisCor onto established codebases with accepted training loops and visualizations. Choice of metrics and length of evaluation seem appropriate.

Clarity: The paper is well written. I did not note any obvious grammatical issues, the organization of the sections was easy to follow, and the use of intuition helped to guide the reader through some of the more theoretical preliminaries.

Relation to Prior Work: Yes. Previous work in function approximation tends to focus on the Bellman backup rather than correcting the underlying data distribution. Prioritized experience replay is a similar concept, but prioritizes based on Bellman error, not value error. Previous work has identified the interaction between data distribution and updates, but has not actually proposed a practical solution. Plenty of appropriate citations to previous efforts around this area.

Reproducibility: Yes

Additional Feedback: Post-rebuttal I had already felt the paper was above the acceptance threshold and appreciated the changes made to improve reader intuition. I am not confident enough in my knowledge of this area to push for an award, but I'm convinced of the significance

[Author Response · NeurIPS 2020]

We thank the reviewers for the detailed comments, suggestions, and a positive assessment of our work. In the final
version of our paper, we shall clarify the details in Section 3 (**R2**), and make intuition in the methods section much
clearer. We will correct for color schemes in all figures (**R1**). We have also made captions of figures cleaner (**R3**).

**R2: concerns regarding Figure 3.** We have added a description of the setup to the paper. $\text{Unif}(s, a)$ is an oracle
distribution where every single $(s, a)$ tuple in the MDP appears in the buffer, exactly the same number of times. This
is of course not achievable in practice, since this data is collected by the policy which might not produce a uniform
distribution and enumerating all state-action pairs in a continuous state (and/or action) MDPs is not possible. This
figure simply argues that for some "oracle" distributions (such as $\text{Unif}(s, a)$), the performance and error reduction in
Q-learning can be much better than the on-policy distribution while retaining the same function approximator and
other details, which provides some evidence that the on-policy distribution is not necessarily optimal in regard to error
reduction with function approximation. On the other hand, without function approximation, on-policy distribution also
performs well (as shown in Fig 3, "Tabular"). That said, $\text{Unif}(s, a)$ is just one (arbitrary) example of a distribution that
performs better than on-policy data. In Fig 5 (left), DisCor actually outperforms $\text{Unif}(s, a)$ on these environments.

**R2: Intuitive explanation of Theorem 4.1.** We have now added a more intuitive discussion for this theorem in the
paper. Intuitively, the optimal distribution assigns higher probability to state-action pairs with high Bellman error
$|Q_k - \mathcal{B}^* Q_{k-1}|$, but only when the overall error $|Q_k - Q^*|$ is minimized. This amounts to minimizing Bellman error
only if the resulting Q-function is going to be closer to $Q^*$. Our tractable approximation (in Sec. 4.2) uses an estimated
error in the target values using $\Delta$ to identify if the resulting Q-function will be closer to $Q^*$. So, intuitively this optimal
distribution up-weights state-action tuples with correct target values, agnostic of the distribution of past policies.

**R2: relation to negative bias in inference on bandit data.** Thank you for pointing us to this literature. We will cite
these papers in the final. These prior works demonstrate how statistical sampling error can induce negative bias in policy
evaluation in bandits. However, the corrective feedback problem we describe is intimately tied to the "bootstrapping" in
ADP updates (not present in bandits) and how on-policy data distributions with function approximation may not be
effective in correcting errors. We will discuss this issue and the connection with statistical error in the paper.

**R3: Lack of corrective feedback in tabular RL.** The issue of absent corrective feedback may indeed arise in tabular
settings with few samples, and we will expand on this discussion in the paper. That said, DisCor is primarily focused
on the case with function approximation, where this problem is particularly exacerbated, as shown in the tree MDP
example (Figure 1) and also occurs in gridworld MDPs (Figure 2 and 3).

**R5: High variance in meta-world.** The reason for high variance on some tasks is likely because learning in different
runs picked up at different times, which is probably because these tasks are especially hard to learn from (as also seen
in the high variance in standard SAC runs). The new runtime with $\Delta$ is 1.3-1.4x of regular SAC.

**R2 and R3: Exact source of problem with function approximation.** To address this, we will add a simple computa-
tional example that illustrates that, even in a simple MDP, error can increase with standard Q-learning but decreases
with DisCor. **Example:** Our example is a 5-state MDP, with the starting state $s_0$ and the terminal state $s_T$ (marked in
gray). Each state has two available actions, $a_0$ and $a_1$, and each action deterministically transits the agent to a state
marked by arrows in Figure 1. A reward of 0.001 is received only when action $a_0$ is chosen at state $s_3$ (else reward is 0).

The Q-function is a linear function over pre-defined features $\phi(s, a)$, i.e., $Q(s, a) = [w_1, w_2]^T \phi(s, a)$, where $\phi(\cdot, a_0) = [1, 1]$ and $\phi(\cdot, a_1) = [1, 1.001]$ (hence features are aliased across states). Computationally, we see that when minimizing Bellman error starting from a Q-function with weights $[w_1, w_2] = [0, 1\text{e-}4]$, under the on-policy distribution of the Boltzmann policy, $\pi(a_0|\cdot) = 0.001, \pi(a_1|\cdot) = 0.999$, in the absence of sampling error (using all transitions but weighted), the error against $Q^*$ still **increases** from 7.177e-3 to 7.179e-3 in one iteration, whereas with DisCor error **decreases** to **5.061e-4**. With uniform the error also decreases, but is larger: 4.776e-3.

44  Figure 1: A simple MDP showing the effect of on-policy
45  distribution and function approximation on learning dynam-
46  ics of ADP algorithms.

48  **Intuition for the example**: The Q-function value error at state-action pairs that will be used as bootstrapping targets
49  for other state-action tuples ($Q(s_0, a_1)$ is used as target for all states with action $a_1$) is high and the state-action pair
50  with correct target value, $(s_3, a_0)$, appears infrequently in the on-policy distribution, since the policy chooses the other
51  action $a_1$ with high probability. Sine the function approximator couples together updates across states and actions,
52  this infrequency of update at $(s_3, a_0)$ and higher frequency of state-action tuples with incorrect targets will update the
53  Q-function approximator towards increasing value error. Thus, minimizing Bellman error can lead to an increase in the
54  error to $Q^*$ (Also shown in **Fig. 2** on a gridworld). We can further generalize this discussion over multiple iterations of
55  learning. This example is a computational version of the tree MDP shown in Figure 1 of the paper **[R2, R5]**.

[Meta-Review · NeurIPS 2020]

The reviewers appreciated the rebuttal that provided some additional insights. Overall a very nice paper with a nicely justified method and very solid experimental evaluation.